# Supermasks in Superposition

**Mitchell Wortsman**[*]
University of Washington

**Vivek Ramanujan**[*]
Allen Institute for AI

**Rosanne Liu**
ML Collective

**Aniruddha Kembhavi**[†]
Allen Institute for AI

**Mohammad Rastegari**
University of Washington

**Jason Yosinski**
ML Collective

**Ali Farhadi**
University of Washington

## Abstract

We present the Supermasks in Superposition (SupSup) model, capable of sequentially learning thousands of tasks without catastrophic forgetting. Our approach uses a randomly initialized, fixed base network and for each task finds a subnetwork (supermask) that achieves good performance. If task identity is given at test time, the correct subnetwork can be retrieved with minimal memory usage. If not provided, SupSup can infer the task using gradient-based optimization to find a linear superposition of learned supermasks which minimizes the output entropy. In practice we find that a single gradient step is often sufficient to identify the correct mask, even among 2500 tasks. We also showcase two promising extensions. First, SupSup models can be trained entirely without task identity information, as they may detect when they are uncertain about new data and allocate an additional supermask for the new training distribution. Finally the entire, growing set of supermasks can be stored in a constant-sized reservoir by implicitly storing them as attractors in a fixed-sized Hopfield network.

## 1  Introduction

Learning many different tasks sequentially without forgetting remains a notable challenge for neural networks [47, 56, 23]. If the weights of a neural network are trained on a new task, performance on previous tasks often degrades substantially [33, 10, 12], a problem known as *catastrophic forgetting*. In this paper, we begin with the observation that catastrophic forgetting cannot occur if the weights of the network remain fixed and random. We leverage this to develop a flexible model capable of learning thousands of tasks: *Supermasks in Superposition* (SupSup). SupSup, diagrammed in Figure 1, is driven by two core ideas: **a)** the expressive power of untrained, randomly weighted subnetworks [57, 39], and **b)** inference of task-identity as a gradient-based optimization problem.

**a) The expressive power of subnetworks**   Neural networks may be overlaid with a binary mask that selectively keeps or removes each connection, producing a subnetwork. The number of possible subnetworks is combinatorial in the number of parameters. Researchers have observed that the number of combinations is large enough that even within randomly weighted neural networks, there exist *supermasks* that create corresponding subnetworks which achieve good performance on complex tasks. Zhou *et al.* [57] and Ramanujan *et al.* [39] present two algorithms for finding these supermasks while keeping the weights of the underlying network fixed and random. SupSup scales to many tasks by finding for each task a supermask atop a shared, untrained network.

**b) Inference of task-identity as an optimization problem**   When task identity is unknown, SupSup can infer task identity to select the correct supermask. Given data from task $j$, we aim

---

[*]Equal contribution. [†]Also affiliated with the University of Washington. Code available at `https://github.com/RAIVNLab/supsup` and correspondence to `{mitchnw,ramanv}@cs.washington.edu`.

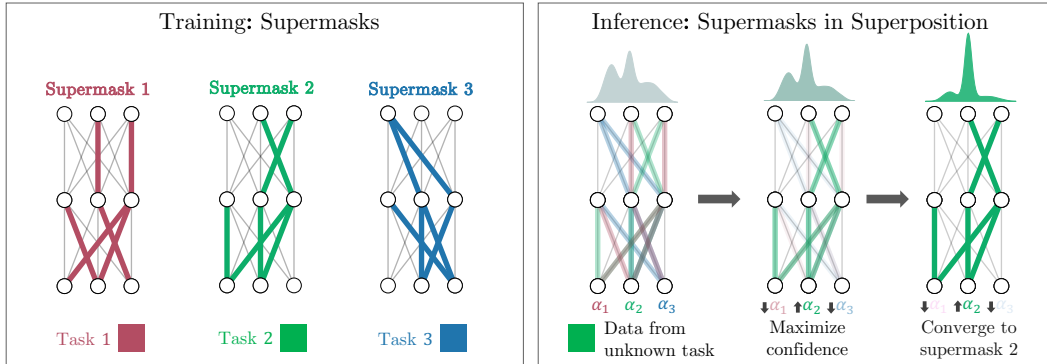

Figure 1: **(left)** During training SupSup learns a separate supermask (subnetwork) for each task. **(right)** At inference time, SupSup can infer task identity by superimposing all supermasks, each weighted by an $\alpha_i$, and using gradients to maximize confidence.

to recover and use the supermask originally trained for task $j$. This supermask should exhibit a confident (*i.e.* low entropy) output distribution when given data from task $j$ [19], so we frame inference of task-identity as an optimization problem—find the convex combination of learned supermasks which minimizes the entropy of the output distribution.

In the rest of the paper we develop and evaluate SupSup via the following contributions:

1. We propose a new taxonomy of continual learning scenarios. We use it to embed and contextualize related work (Section 2).

2. When task identity (ID) is provided during train and test (later dubbed GG), SupSup is a natural extension of Mallya *et al.* [30]. By using a randomly weighted backbone and controlling mask sparsity, SupSup surpasses recent baselines on SplitImageNet [51] while requiring less storage and time costs (Section 3.2).

3. When task ID is provided during train but not test (later dubbed GN), SupSup outperforms recent methods that require task ID [26, 23, 4], scaling to 2500 permutations of MNIST without forgetting. For these uniform tasks, ID can be inferred with a single gradient computation (Section 3.3).

4. When task identities are not provided at all (later dubbed NNs), SupSup can even infer task boundaries and allocate new supermasks as needed (Section 3.4).

5. We introduce an extension to the basic SupSup algorithm that stores supermasks implicitly as attractors in a fixed-size Hopfield network [20] (Section 3.5).

6. Finally, we empirically show that the simple trick of adding *superfluous neurons* results in more accurate task inference (Section 3.6).

## 2   Continual Learning Scenarios and Related Work

In continual learning, a model aims to solve a number of tasks sequentially [47, 56] without catastrophic forgetting [10, 23, 33]. Although numerous approaches have been proposed in the context of continual learning, there lacks a convention of scenarios in which methods are trained and evaluated [49]. The key identifiers of scenarios include: **1)** whether task identity is provided during training, **2)** provided during inference, **3)** whether class labels are shared during evaluation, and **4)** whether the overall task space is discrete or continuous. This results in an exhaustive set of 16 possibilities, many of which are invalid or uninteresting. For example, if task identity is never provided in training, providing it in inference is no longer helpful. To that end, we highlight four applicable scenarios, each with a further breakdown of discrete vs. continuous, when applicable, as shown in Table 1.

We decompose continual learning scenarios via a three-letter taxonomy that explicitly addresses the three most critical scenario variations. The first two letters specify whether task identity is given during training (G if given, N if not) and during inference (G if given, N if not). The third letter specifies a subtle but important distinction: whether labels are shared (s) across tasks or not (u). In the unshared case, the model must predict both the correct task ID and the correct class within that

Table 1: Overview of different Continual Learning scenarios. We suggest scenario names that provide an intuitive understanding of the variations in training, inference, and evaluation, while allowing a full coverage of the scenarios previously defined in [49] and [55]. See text for more complete description.

| Scenario | Description | Task space discreet or continuous? | Example methods / task names used |
|---|---|---|---|
| GG | Task **G**iven during train and **G**iven during inference | Either | PNN [42], BatchE [51], PSP [4], "Task learning" [55], "Task-IL" [49] |
| GNs | Task **G**iven during train, **N**ot inference; **s**hared labels | Either | EWC [23], SI [54], "Domain learning" [55], "Domain-IL" [49] |
| GNu | Task **G**iven during train, **N**ot inference; **u**nshared labels | Discrete only | "Class learning" [55], "Class-IL" [49] |
| NNs | Task **N**ot given during train **N**or inference; **s**hared labels | Either | BGD, "Continuous/discrete task agnostic learning" [55] |

task. In the shared case, the model need only predict the correct, shared label across tasks, so it need not represent or predict which task the data came from. For example, when learning 5 permutations of MNIST in the GN scenario (task IDs given during train but not test), a shared label GNs scenario will evaluate the model on the correct predicted label across 10 possibilities, while in the unshared GNu case the model must predict across 50 possibilities, a more difficult problem.

A full expansion of possibilities entails both GGs and GGu, but as s and u describe only model *evaluation*, any model capable of predicting shared labels can predict unshared equally well using the provided task ID at test time. Thus these cases are equivalent, and we designate both GG. Moreover, the NNu scenario is invalid because unseen labels signal the presence of a new task (the "labels trick" in [55]), making the scenario actually GNu, and so we consider only the shared label case NNs.

We leave out the discrete vs. continuous distinction as most research efforts operate within one framework or the other, and the taxonomy applies equivalently to discrete domains with integer "Task IDs" as to continue domains with "Task Embedding" or "Task Context" vectors. The remainder of this paper follows the majority of extant literature in focusing on the case with discrete task boundaries (see e.g. [55] for progress in the continuous scenario). Equipped with this taxonomy, we review three existing approaches for continual learning.

**(1) Regularization based methods** Methods like Elastic Weight Consolidation (EWC) [23] and Synaptic Intelligence (SI) [54] penalize the movement of parameters that are important for solving previous tasks in order to mitigate catastrophic forgetting. Measures of parameter importance vary; e.g. EWC uses the Fisher Information matrix [36]. These methods operate in the GNs scenario (Table 1). Regularization approaches ameliorate but do not exactly eliminate catastrophic forgetting.

**(2) Using exemplars, replay, or generative models** These methods aim to explicitly or implicitly (with generative models) capture data from previous tasks. For instance, [40] performs classification based on the nearest-mean-of-examplars in a feature space. Additionally, [27, 3] prevent the model from increasing loss on examples from previous tasks while [41] and [45] respectively use memory buffers and generative models to replay past data. Exact replay of the entire dataset can trivially eliminate catastrophic forgetting but at great time and memory cost. Generative approaches can reduce catastrophic forgetting, but generators are also susceptible to forgetting. Recently, [50] successfully mitigate this obstacle by parameterizing a generator with a hypernetwork [15].

**(3) Task-specific model components** Instead of modifying the learning objective or replaying data, various methods [42, 53, 31, 30, 32, 52, 4, 11, 51] use different model components for different tasks. In Progressive Neural Networks (PNN), Dynamically Expandable Networks (DEN), and Reinforced Continual Learning (RCL) [42, 53, 52], the model is expanded for each new task. More efficiently, [32] fixes the network size and randomly assigns which nodes are active for a given task. In [31, 11], the weights of disjoint subnetworks are trained for each new task. Instead of learning the weights of the subnetwork, for each new task Mallya *et al.* [30] learn a binary mask that is applied to a network pretrained on ImageNet. Recently, Cheung *et al.* [4] superimpose many models into one by using different (and nearly orthogonal) contexts for each task. The task parameters can then be effectively retrieved using the correct task context. Finally, BatchE [51] learns a shared weight matrix on the first task and learn only a rank-one elementwise scaling matrix for each subsequent task.

Our method falls into this final approach (3) as it introduces task-specific supermasks. However, while all other methods in this category are limited to the GG scenario, SupSup can be used to achieve compelling performance in *all four scenarios*. We compare primarily with BatchE [51] and Parameter Superposition (abbreviated PSP) [4] as they are recent and performative. BatchE requires very few additional parameters for each new task while achieving comparable performance to PNN and scaling to SplitImagenet. Moreover, PSP outperforms regularization based approaches like SI [54]. However,

| Algorithm | Avg Top 1 Accuracy (%) | Bytes |
|---|---|---|
| Upper Bound | 92.55 | 10222.81M |
| SupSup (GG) | 89.58 | 195.18M |
|  | 88.68 | 100.98M |
|  | 86.37 | 65.50M |
| BatchE (GG) | 81.50 | 124.99M |
| Single Model | - | 102.23M |

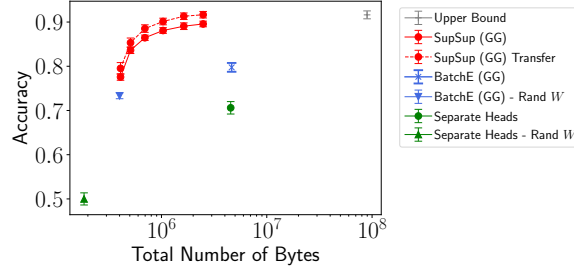

Figure 2: **(left) SplitImagenet** performance in Scenario GG. SupSup approaches upper bound performance with significantly fewer bytes. **(right) SplitCIFAR100** performance in Scenario GG shown as mean and standard deviation over 5 seed and splits. SupSup outperforms similar size baselines and benefits from *transfer*.

both BatchE [51] and PSP [4] require task identity to use task-specific weights, so they can only operate in the GG setting.

## 3 Methods

In this section, we detail how SupSup leverages supermasks to learn thousands of sequential tasks without forgetting. We begin with easier settings where task identity is given and gradually move to more challenging scenarios where task identity is unavailable.

### 3.1 Preliminaries

In a standard $\ell$-way classification task, inputs $\mathbf{x}$ are mapped to a distribution $\mathbf{p}$ over output neurons $\{1, ..., \ell\}$. We consider the general case where $\mathbf{p} = f(\mathbf{x}, W)$ for a neural network $f$ parameterized by $W$ and trained with a cross-entropy loss. In continual learning classification settings we have $k$ different $\ell$-way classification tasks and the input size remains constant across tasks[2].

Zhou *et al.* [57] demonstrate that a trained binary mask (supermask) $M$ can be applied to a randomly weighted neural network, resulting in a subnetwork with good performance. As further explored by Ramanujan *et al.* [39], supermasks can be trained at similar compute cost to training weights while achieving performance competitive with weight training.

With supermasks, outputs are given by $\mathbf{p} = f(\mathbf{x}, W \odot M)$ where $\odot$ denotes an elementwise product. $W$ is kept frozen at its initialization: bias terms are $\mathbf{0}$ and other parameters in $W$ are $\pm c$ with equal probability and $c$ is the standard deviation of the corresponding Kaiming normal distribution [17]. This initialization is referred to as *signed Kaiming constant* by [39] and the constant $c$ may be different for each layer. For completeness we detail the Edge-Popup algorithm for training supermasks [39] in Section E of the appendix.

### 3.2 Scenario GG: Task Identity Information Given During Train and Inference

When task identity is known during training we can learn a binary mask $M^i$ per task. $M^i$ are the only parameters learned as the weights remain fixed. Given data from task $i$, outputs are computed as

$$\mathbf{p} = f(\mathbf{x}, W \odot M^i) \tag{1}$$

For each new task we can either initialize a new supermask randomly, or use a running mean of all supermasks learned so far. During inference for task $i$ we then use $M^i$. Figure 2 illustrates that in this scenario SupSup outperforms a number of baselines in accuracy on both SplitCIFAR100 and SplitImageNet while requiring fewer bytes to store. Experiment details are in Section 4.1.

### 3.3 Scenarios GNs & GNu : Task Identity Information Given During Train Only

We now consider the case where input data comes from task $j$, but this task information is unknown to the model at inference time. During training we proceed exactly as in Scenario GG, obtaining $k$

learned supermasks. During inference, we aim to infer task identity—correctly detect that the data belongs to task $j$—and select the corresponding supermask $M^j$.

The SupSup procedure for task ID inference is as follows: first we associate each of the $k$ learned supermasks $M^i$ with an coefficient $\alpha_i \in [0, 1]$, initially set to $1/k$. Each $\alpha_i$ can be interpreted as the "belief" that supermask $M^i$ is the correct mask (equivalently the belief that the current unknown task is task $i$). The model's output is then be computed with a weighted superposition of all learned masks:

$$\mathbf{p}(\alpha) = f\left(\mathbf{x}, W \odot \left(\sum_{i=1}^{k} \alpha_i M^i\right)\right). \tag{2}$$

The correct mask $M^j$ should produce a confident, low-entropy output [19]. Therefore, to recover the correct mask we find the coefficients $\alpha$ which minimize the output entropy $\mathcal{H}$ of $\mathbf{p}(\alpha)$. One option is to perform gradient descent on $\alpha$ via

$$\alpha \leftarrow \alpha - \eta \nabla_\alpha \mathcal{H}\left(\mathbf{p}\left(\alpha\right)\right) \tag{3}$$

where $\eta$ is the step size, and $\alpha$s are re-normalized to sum to one after each update. Another option is to try each mask individually and pick the one with the lowest entropy output requiring $k$ forward passes. However, we want an optimization method with fixed sub-linear run time (w.r.t. the number of tasks $k$) which leads $\alpha$ to a corner of the probability simplex — *i.e.* $\alpha$ is 0 everywhere except for a single 1. We can then take the nonzero index to be the inferred task. To this end we consider the **One-Shot** and **Binary** algorithms.

**One-Shot:** The task is inferred using a single gradient. Specifically, the inferred task is given by

$$\arg\max_i \left(-\frac{\partial \mathcal{H}\left(\mathbf{p}\left(\alpha\right)\right)}{\partial \alpha_i}\right) \tag{4}$$

as entropy is decreasing maximally in this coordinate. This algorithms corresponds to one step of the Frank-Wolfe algorithm [7], or one-step of gradient descent followed by softmax re-normalization with the step size $\eta$ approaching $\infty$. Unless noted otherwise, $\mathbf{x}$ is a single image and not a batch.

**Binary:** Resembling binary search, we infer task identity using an algorithm with $\log k$ steps. At each step we rule out half the tasks—the tasks corresponding to entries in the bottom half of $-\nabla_\alpha \mathcal{H}\left(\mathbf{p}\left(\alpha\right)\right)$. These are the coordinates in which entropy is minimally decreasing. A task $i$ is ruled out by setting $\alpha_i$ to zero and at each step we re-normalize the remaining entries in $\alpha$ so that they sum to one. Pseudo-code for both algorithms may be found in Section A of the appendix.

Once the task is inferred the corresponding mask can be used as in Equation 1 to obtain class probabilities $\mathbf{p}$. In both Scenario GNs and GNu the class probabilities $\mathbf{p}$ are returned. In GNu, $\mathbf{p}$ forms a distribution over the classes corresponding to the inferred task. Experiments solving thousands of tasks are detailed in Section 4.2.

### 3.4 Scenario NNs: No Task Identity During Training or Inference

Task inference algorithms from Scenario GN enable the extension of SupSup to Scenario NNs, where task identity is entirely unknown (even during training). If SupSup is uncertain about the current task identity, it is likely that the data do not belong to any task seen so far. When this occurs a new supermask is allocated, and $k$ (the number of tasks learned so far) is incremented.

We consider the **One-Shot** algorithm and say that SupSup is uncertain when performing task identity inference if $\nu = \text{softmax}\left(-\nabla_\alpha \mathcal{H}\left(\mathbf{p}\left(\alpha\right)\right)\right)$ is approximately uniform. Specifically, if $k \max_i \nu_i < 1 + \epsilon$ a new mask is allocated and $k$ is incremented. Otherwise mask $\arg\max_i \nu_i$ is used, which corresponds to Equation 4. We conduct experiments on learning up to 2500 tasks entirely without any task information, detailed in Section 4.3. Figure 4 shows that SupSup in Scenario NNs achieves comparable performance even to Scenario GNu.

### 3.5 Beyond Linear Memory Dependence

Hopfield networks [20] implicitly encode a series of binary strings $\mathbf{z}^i \in \{-1, 1\}^d$ with an associated energy function $E_\Psi(\mathbf{z}) = \sum_{uv} \Psi_{uv} \mathbf{z}_u \mathbf{z}_v$. Each $\mathbf{z}^i$ is a minima of $E_\Psi$, and can be recovered with gradient descent. $\Psi \in \mathbb{R}^{d \times d}$ is initially $\mathbf{0}$, and to encode a new string $z^i$, $\Psi \leftarrow \Psi + \frac{1}{d}\mathbf{z}^i {\mathbf{z}^i}^\top$.

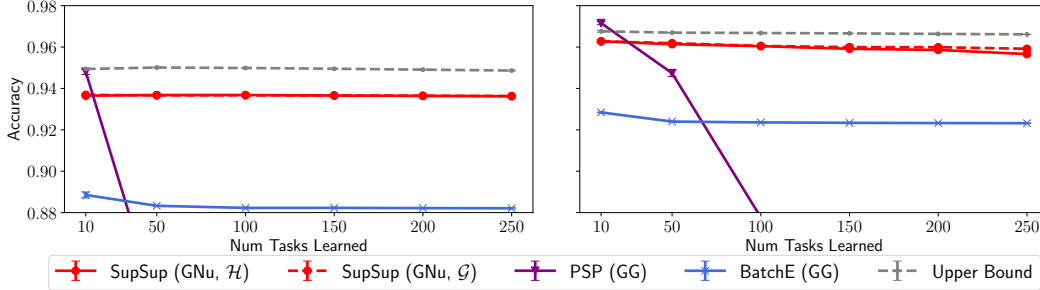

Figure 3: Using **One-Shot** to infer task identity, SupSup outperforms methods with access to task identity. Results shown for PermutedMNIST with LeNet 300-100 **(left)** and FC 1024-1024 **(right)**.

We now consider implicitly encoding the masks in a fixed-size Hopfield network $\Psi$ for Scenario GNu. For a new task $i$ a new mask is learned. After training on task $i$, this mask will be stored as an attractor in a fixed size Hopfield network. Given new data during inference we perform gradient descent on the Hopfield energy $E_\Psi$ with the output entropy $\mathcal{H}$ to learn a new mask $\mathbf{m}$. Minimizing $E_\Psi$ will hopefully push $\mathbf{m}$ towards a mask learned during training while $\mathcal{H}$ will push $\mathbf{m}$ to be the correct mask. As $\Psi$ is quadratic in mask size, we will not mask the parameters $W$. Instead we mask the output of every layer except the last, *e.g.* a network with one hidden layer and mask $\mathbf{m}$ is given by

$$f(\mathbf{x}, \mathbf{m}, W) = \texttt{softmax}\left(W_2^\top\left(\mathbf{m} \odot \sigma\left(W_1^\top \mathbf{x}\right)\right)\right) \tag{5}$$

for nonlinearity $\sigma$. The Hopfield network will then be a similar size as the base neural network. We refer to this method as HopSupSup and provide additional details in Section B.

### 3.6 Superfluous Neurons & an Entropy Alternative

Similar to previous methods [49], HopSupSup requires $\ell k$ output neurons in Scenario GNu. SupSup, however, is performing $\ell k$-way classification without $\ell k$ output neurons. Given data during inference **1)** the task is inferred and **2)** the corresponding mask is used to obtain outputs $\mathbf{p}$. The class probabilities $\mathbf{p}$ correspond to the classes for the inferred task, effectively reusing the neurons in the final layer.

SupSup could use an output size of $\ell$, though we find in practice that it helps significantly to add extra neurons to the final layer. Specifically we consider outputs $\mathbf{p} \in \mathbb{R}^s$ and refer to the neurons $\{\ell+1, ..., s\}$ as superfluous neurons (s-neurons). The standard cross-entropy loss will push the values of s-neurons down throughout training. Accordingly, we consider an objective $\mathcal{G}$ which encourages the s-neurons to have large negative values and can be used as an alternative to entropy in Equation 4. Given data from task $j$, mask $M^j$ will minimize the values of the s-neurons as it was trained to do. Other masks were also trained to minimize the values of the s-neurons, but not for data from task $j$. In Lemma 1 of Section I we provide the exact form of $\mathcal{G}$ in code ($\mathcal{G} = \texttt{logsumexp}\,(\mathbf{p})$ with masked gradients for $\mathbf{p}_1, ..., \mathbf{p}_\ell$) and offer an alternative perspective on why $\mathcal{G}$ is effective — the gradient of $\mathcal{G}$ for all s-neurons exactly mirrors the gradient from the supervised training loss.

## 4 Experiments

### 4.1 Scenario GG: Task Identity Information Given During Train and Inference

**Datasets, Models & Training**  In this experiment we validate the performance of SupSup on SplitCIFAR100 and SplitImageNet. Following Wen *et al.* [51], SplitCIFAR100 randomly partitions CIFAR100 [24] into 20 different 5-way classification problems. Similarly, SplitImageNet randomly splits the ImageNet [5] dataset into 100 different 10-way classification tasks. Following [51] we use a ResNet-18 with fewer channels for SplitCIFAR100 and a standard ResNet-50 [18] for SplitImageNet. The Edge-Popup algorithm from [39] is used to obtain supermasks for various sparsities with a layer-wise budget from [35]. We either initialize each new mask randomly (as in [39]) or use a running mean of all previous learned masks. This simple method of "Transfer" works very well, as illustrated by Figure 2. Additional training details and hyperparameters are provided in Section D.

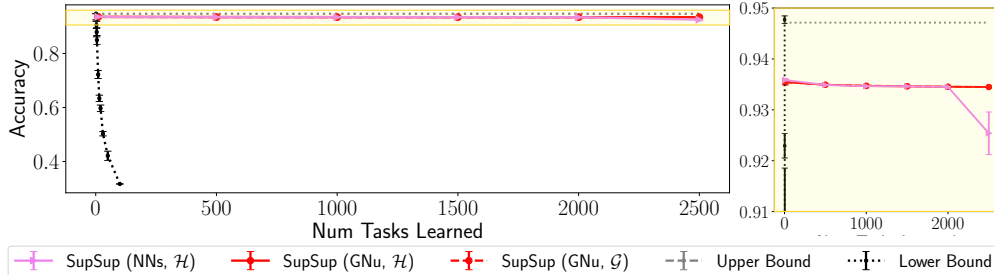

Figure 4: Learning 2500 tasks and inferring task identity using the **One-Shot** algorithm. Results for both the GNu and NNs scenarios with the LeNet 300-100 model using output size 500.

**Computation** In Scenario GG, the primary advantage of SupSup from Mallya *et al.* [31] or Wen *et al.* [51] is that SupSup does not require the base model $W$ to be stored. Since $W$ is random it suffices to store only the random seed. For a fair comparison we also train BatchE [51] with random weights. The sparse supermasks are stored in the standard `scipy.sparse.csc`[3] format with 16 bit integers. Moreover, SupSup requires minimal overhead in terms of forwards pass compute. Elementwise product by a binary mask can be implemented via memory access, *i.e.* selecting indices. Modern GPUs have very high memory bandwidth so the time cost of this operation is small with respect to the time of a forward pass. In particular, on a 1080 Ti this operation requires $\sim 1\%$ of the forward pass time for a ResNet-50, less than the overhead of BatchE (computation in Section D).

**Baselines** In Figure 2, for "Separate Heads" we train different heads for each task using a *trunk* (all layers except the final layer) trained on the first task. In contrast "Separate Heads - Rand W" uses a random trunk. BatchE results are given with the trunk trained on the first task (as in [51]) and random weights $W$. For "Upper Bound", individual models are trained for each task. Furthermore, the trunk for task $i$ is trained on tasks $1, ..., i$. For "Lower Bound" a shared trunk of the network is trained continuously and a separate head is trained for each task. Since catastrophic forgetting occurs we omit "Lower Bound" from Figure 2 (the SplitCIFAR100 accuracy is 24.5%).

## 4.2 Scenarios GNs & GNu: Task Identity Information Given During Train Only

Our solutions for GNs and GNu are very similar. Because GNu is strictly more difficult, we focus on only evaluating in Scenario GNu. For relevant figures we provide a corresponding table in Section H.

**Datasets** Experiments are conducted on PermutedMNIST, RotatedMNIST, and SplitMNIST. For PermutedMNIST [23], new tasks are created with a fixed random permutation of the pixels of MNIST. For RotatedMNIST, images are rotated by 10 degrees to form a new task with 36 tasks in total (similar to [4]). Finally SplitMNIST partitions MNIST into 5 different 2-way classification tasks, each containing consecutive classes from the original dataset.

**Training** We consider two architectures: **1)** a fully connected network with two hidden layers of size 1024 (denoted FC 1024-1024 and used in [4]) **2)** the LeNet 300-100 architecture [25] as used in [8, 6]. For each task we train for 1000 batches of size 128 using the RMSProp optimizer [48] with learning rate 0.0001 which follows the hyperparameters of [4]. Supermasks are found using the algorithm of Mallya *et al.* [31] with threshold value 0. However, we initialize the real valued "scores" with Kaiming uniform as in [39]. Training the mask is not a focus of this work, we choose this method as it is fast and we are not concerned about controlling mask sparsity as in Section 4.1.

**Evaluation** At test time we perform inference of task identity once for each batch. If task is not inferred correctly then accuracy is 0 for the batch. Unless noted otherwise we showcase results for the most challenging scenario — when the task identity is inferred using a single image. We use "Full Batch" to indicate that all 128 images are used to infer task identity. Moreover, we experiment with both the the entropy $\mathcal{H}$ and $\mathcal{G}$ (Section 3.6) objectives to perform task identity inference.

**Results** Figure 4 illustrates that SupSup is able to sequentially learn 2500 permutations of MNIST— SupSup succeeds in performing 25,000-way classification. This experiment is conducted with the **One-Shot** algorithm (requiring one gradient computation) using single images to infer task identity. The same trends hold in Figure 3, where SupSup outperforms methods which operate in Scenario GG

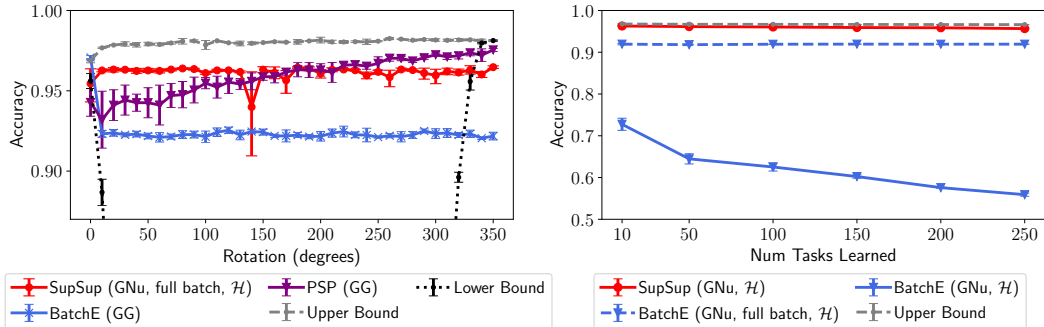

Figure 5: **(left)** Testing the FC 1024-1024 model on RotatedMNIST. SupSup uses **Binary** to infer task identity with a full batch as tasks are similar (differing by only 10 degrees). **(right)** The **One-Shot** algorithm can be used to infer task identity for BatchE [51]. Experiment conducted with FC 1024-1024 on PermutedMNIST using an output size of 500, shown as mean and stddev over 3 runs.

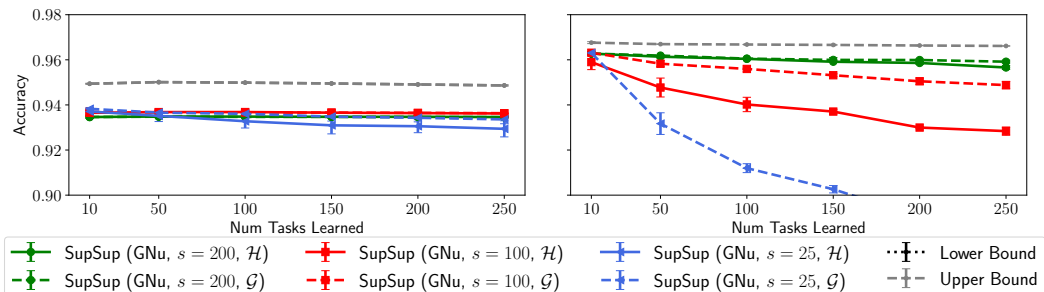

Figure 6: The effect of output size $s$ on SupSup performance using the **One-Shot** algorithm. Results shown for PermutedMNIST with LeNet 300-100 **(left)** and FC 1024-1024 **(right)**.

by using the **One-Shot** algorithm to infer task identity. In Figure 3, output sizes of 100 and 500 are respectively used for LeNet 300-100 and FC 1024-1024. The left hand side of Figure 5 illustrates that SupSup is able to infer task identity even when tasks are similar—SupSup is able to distinguish between rotations of 10 degrees. Since this is a more challenging problem, we use a full batch and the **Binary** algorithm to perform task identity inference. Figure 7 (appendix) shows that for HopSupSup on SplitMNIST, the new mask $\mathbf{m}$ converges to the correct supermask in $< 30$ gradient steps.

**Baselines & Ablations** Figure 5 (left) shows that even in Scenario GNu, SupSup is able to outperform PSP [4] and BatchE [51] in Scenario GG—methods using task identity. We compare SupSup in GNu with methods in this strictly easier scenario as they are more competitive. For instance, [49] considers sequential learning problems with only 5-10 tasks. SupSup, after sequentially learning 250 permutations of MNIST, outperforms all non-replay methods from [3] in the GNu scenario after they have learned only 10 permutations of MNIST with a similar network. In GNu, Online EWC achieves 33.88% & SI achieves 29.31% on 10 permutations of MNIST [49] while SupSup achieves 94.91% accuracy after 250 permutations (see Table 5 in [49] vs. Table 7).

In Figure 5 (right) we equip BatchE with task inference using our **One-Shot** algorithm. Instead of attaching a weight $\alpha_i$ to each supermask, we attach a weight $\alpha_i$ to each rank-one matrix [51]. Moreover, in Section C of the appendix we augment BatchE to perform task-inference using large batch sizes. "Upper Bound" and "Lower Bound" are the same as in Section 4.1. Moreover, Figure 6 illustrates the importance of output size. Further investigation of this phenomena is provided by Section 3.6 and Lemma 1 of Section I.

### 4.3 Scenario NNs: No Task Identity During Training or Inference

For the NNs Scenario we consider PermutedMNIST and train on each task for 1000 batches (the model does not have access to this iteration number). Every 100 batches the model must choose to allocate a new mask or pick an existing mask using the criteria from Section 3.4 ($\epsilon = 2^{-3}$). Figure 4 illustrates that without access to any task identity (even during training) SupSup is able to learn thousands of tasks. However, a final dip is observed as a budget of 2500 supermasks total is enforced.

# 5    Conclusion

Supermasks in Superposition (SupSup) is a flexible and compelling model applicable to a wide range of scenarios in Continual Learning. SupSup leverages the power of subnetworks [57, 39, 31], and gradient-based optimization to infer task identity when unknown. SupSup achieves state-of-the-art performance on SplitImageNet when given task identity, and performs well on thousands of permutations and almost indiscernible rotations of MNIST without any task information.

We observe limitations in applying SupSup with task identity inference to non-uniform and more challenging problems. Task inference fails when models are not well calibrated—are overly confident for the wrong task. As future work, we hope to explore automatic task inference with more calibrated models [14], as well as circumventing calibration challenges by using optimization objectives such as self-supervision [16] and energy based models [13]. In doing so, we hope to tackle large-scale problems in Scenarios GN and NNs.

## Broader Impact

A goal of continual learning is to solve many tasks with a single model. However, it is not exactly clear what qualifies as a *single model*. Therefore, a concrete objective has become to learn many tasks as efficiently as possible. We believe that SupSup is a useful step in this direction. However, there are consequences to more efficient models, both positive and negative.

We begin with the positive consequences:

- Efficient models require less compute, and are therefore less harmful for the environment then learning one model per task [44]. This is especially true if models are able to leverage information from past tasks, and training on new tasks is then faster.

- Efficient models may be run on the end device. This helps to preserve privacy as a user's data does not have to be sent to the cloud for computation.

- If models are more efficient then large scale research is not limited to wealthier institutions. These institutions are more likely in privileged parts of the world and may be ignorant of problems facing developing nations. Moreover, privileged institutions may not be a representative sample of the research community.

We would also like to highlight and discuss the negative consequences of models which can efficiently learn many tasks, and efficient models in general. When models are more efficient, they are also more available and less subject to regularization and study as a result. For instance, when a high-impact model is released by an institution it will hopefully be accompanied by a Model Card [34] analyzing the bias and intended use of the model. By contrast, if anyone is able to train a powerful model this may no longer be the case, resulting in a proliferation of models with harmful biases or intended use. Taking the United States for instance, bias can be harmful as models show disproportionately more errors for already marginalized groups [2], furthering existing and deeply rooted structural racism.

### Acknowledgments

We thank Gabriel Ilharco Magalhães and Sarah Pratt for helpful comments. For valuable conversations we also thank Tim Dettmers, Kiana Ehsani, Ana Marasović, Suchin Gururangan, Zoe Steine-Hanson, Connor Shorten, Samir Yitzhak Gadre, Samuel McKinney and Kishanee Haththotuwegama. This work is in part supported by NSF IIS 1652052, IIS 17303166, DARPA N66001-19-2-4031, DARPA W911NF-15-1-0543 and gifts from Allen Institute for Artificial Intelligence. Additional revenues: co-authors had employment with the Allen Institute for AI.

## Footnotes

[2]In practice the tasks do not all need to be $\ell$-way — output layers can be padded until all have the same size.

[3]`https://docs.scipy.org/doc/scipy/reference/sparse.html`

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
