[Supplementary Material]

**Algorithm 1** One-Shot$(f, \mathbf{x}, W, k, \{M^i\}_{i=1}^k, \mathcal{H})$

1: $\alpha \leftarrow \begin{bmatrix} \frac{1}{k} & \frac{1}{k} & ... & \frac{1}{k} \end{bmatrix}$         ▷ Initialize $\alpha$
2: $\mathbf{p} \leftarrow f\left(\mathbf{x}, W \odot \left(\sum_{i=1}^k \alpha_i M^i\right)\right)$        ▷ Superimposed output
3: **return** $\arg\max_i \left(-\frac{\partial \mathcal{H}(\mathbf{p})}{\partial \alpha_i}\right)$    ▷ Return coordinate for which objective maximally decreasing

---

**Algorithm 2** Binary$(f, \mathbf{x}, W, k, \{M^i\}_{i=1}^k, \mathcal{H})$

1: $\alpha \leftarrow \begin{bmatrix} \frac{1}{k} & \frac{1}{k} & ... & \frac{1}{k} \end{bmatrix}$         ▷ Initialize $\alpha$
2: **while** $\|\alpha\|_0 > 1$ **do**        ▷ Iterate until $\alpha$ has a single nonzero entry
3:     $\mathbf{p} \leftarrow f\left(\mathbf{x}, W \odot \left(\sum_{i=1}^k \alpha_i M^i\right)\right)$       ▷ Superimposed output
4:     $g \leftarrow -\nabla_\alpha \mathcal{H}(\mathbf{p})$        ▷ Gradient of objective
5:     **for** $i \in \{1, ..., k\}$ **do**        ▷ In code this **for** loop is vectorized
6:        **if** $g_i \leq \mathbf{median}(g)$ **then**
7:           $\alpha_i \leftarrow 0$       ▷ Zero out $\alpha_i$ for which objective minimally decreasing
8:     $\alpha \leftarrow \alpha / \|\alpha\|_1$        ▷ Re-normalize $\alpha$ to sum to 1
9: **return** $\arg\max_i \alpha_i$

## A Algorithm pseudo-code

Algorithms 1 and 2 respectively provide pseudo-code for the **One-Shot** and **Binary** algorithms detailed in Section 3.3. Both aim to infer the task $j \in \{1, ..., k\}$ associated with input data $\mathbf{x}$ by minimizing the objective $\mathcal{H}$.

## B Extended Details for HopSupSup

This section provides further details and experiments for HopSupSup (introduced in Section 3.5). HopSupSup provides a method for storing the growing set of supermasks in a fixed size reservoir instead of explicitly storing each mask.

### B.1 Training

Recall that HopSupSup operates in Scenario GNu and so task identity is known during training. Instead of explicitly storing each mask, we will instead store two fixed sized variables $\Psi$ and $\mu$ which are both initially $\mathbf{0}$. The weights of the Hopfield network are $\Psi$ and $\mu$ stores a running mean of all masks learned so far. For a new task $k$ we use the same algorithm as in Section 4.2 to learn a binary mask $\mathbf{m}^i$ which performs well for task $k$. Since Hopfield networks consider binary strings in $\{-1, 1\}^d$ and we use masks $\mathbf{m}^i \in \{0, 1\}^d$ we will consider $\mathbf{z}^k = 2\mathbf{m}^k - 1$. In practice we then update $\Psi$ and $\mu$ as

$$\Psi \leftarrow \Psi + \frac{1}{d}\left(\mathbf{z}^k \mathbf{z}^{k\top} - \mathbf{z}^k \left(\Psi \mathbf{z}^k\right)^\top - \left(\Psi \mathbf{z}^k\right)\mathbf{z}^{k\top} - \text{Id}\right), \qquad \mu \leftarrow \frac{k-1}{k}\mu + \frac{1}{k}\mathbf{z}^k \qquad (6)$$

where Id is the identity matrix. This update rule for $\Psi$ is referred to as the Storkey learning rule [46] and is more expressive than the alternative—the Hebbian rule $\Psi \leftarrow \Psi + \frac{1}{d}\mathbf{z}^k \mathbf{z}^{k\top}$ [20] provided for brevity in Section 3.3. With either update rules the learned $\mathbf{z}^i$ will be a minimizer of the Hopfield energy $E_\Psi(\mathbf{z}) = \sum_{uv} \Psi_{uv} \mathbf{z}_u \mathbf{z}_v$.

### B.2 Inference

During inference we receive data $\mathbf{x}$ from some task $j$, but this task information is not given to the model. HopSupSup first initializes a new binary string $\mathbf{z}$ with $\mu$. Next, HopSupSup uses gradient descent to minimize the Hopfield energy in conjunction with the output entropy using mask $\mathbf{m} = \frac{1}{2}\mathbf{z} + 1$, a process we refer to as *Hopfield Recovery*. Minimizing the energy will hopefully push $\mathbf{m}$ (equivalently $\mathbf{z}$) towards a mask learned during training and minimizing the entropy will hopefully

Figure 7: During *Hopfield Recovery* the new mask $\mathbf{m}$ converges to the correct mask learned during training. Note that $\mathbf{m}^i$ denotes the mask learned for task $i$.

push $\mathbf{m}$ towards the correct mask $\mathbf{m}^j$. We may then use the recovered mask to compute the network output.

In practice we use one pass through the evaluation set (with batch size 64, requiring $T \approx 30$ steps) to recover a mask and another to perform evaluation with the recovered mask. When recovering the mask we gradually increase the strength of the Hopfield term and decrease the strength of the entropy term. Otherwise the Hopfield term initially pulls $\mathbf{z}$ in the wrong direction or the final $\mathbf{z}$ does not lie at a minimum of $E_\Psi$. For step $t \in \{1, ..., T\}$, and constant $\gamma$ we use the objective $\mathcal{J}$ as

$$\mathcal{J}(\mathbf{z}, t) = \frac{\gamma t}{T} E_\Psi(\mathbf{z}) + \left(1 - \frac{t}{T}\right) \mathcal{H}(\mathbf{p}) \tag{7}$$

where $\mathbf{p}$ denotes the output using mask $\mathbf{m} = \frac{1}{2}\mathbf{z} + 1$.

Figure 7 illustrates that after approximately 30 steps of gradient descent on $\mathbf{z}$ using objective $\mathcal{J}$, the mask $\mathbf{m} = \frac{1}{2}\mathbf{z} + 1$ converges to the correct mask learned during training. This experiment is conducted for 20 different random seeds on SplitMNIST (see Section 4.2) training for 1 epoch per task. Evaluation with the recovered mask for each seed is then given by Figure 8. As expected, when the correct mask is successfully recovered, accuracy matches directly using the correct mask. For hyperparameters we set $\gamma = 1.5 \cdot 10^{-3}$ and perform gradient descent during Hopfield recovery with learning rate $0.5 \cdot 10^3$, momentum 0.9, and weight decay $10^{-4}$.

### B.3 Network Architecture

Let BN denote non-affine batch normalization [21], *i.e.* batch normalization with no learned parameters. Also recall that we are masking layer outputs instead of weights, and the weights still remain fixed (see Section 3.5). Therefore, with mask $\mathbf{m} = (\mathbf{m}_1, \mathbf{m}_2)$ and weights $W = (W_1, W_2, W_3)$ we compute outputs as

$$f(\mathbf{x}, \mathbf{m}, W) = \texttt{softmax}\left(W_3^\top \sigma\left(\mathbf{m}_2 \odot \texttt{BN}\left(W_2^\top \sigma\left(\mathbf{m}_1 \odot \texttt{BN}\left(W_1^\top \mathbf{x}\right)\right)\right)\right)\right) \tag{8}$$

where $\sigma$ denotes the Swish nonlinearity [38]. Without masking or normalization $f$ is a fully connected network with two hidden layers of size 2048. We also note that HopSupSup requires 10 output neurons for SplitMNIST in Scenario GNu, and the composition of non-affine batch normalization with a binary mask was inspired by BatchNets [9].

## C  Augmenting BatchE For Scnario GNu

In Section 4.2 we demonstrate that BatchE [51] is able to infer task identity using the **One-Shot** algorithm. In this section we show that, equipped with $\mathcal{H}$ from Section 3, BatchE can also infer task identity by using a large batch size. We refer to this method as Augmented BatchE (ABatchE).

Figure 8: Evaluating (with 20 random seeds) on SplitM-NIST after finding a mask with *Hopfield Recovery*. Average accuracy is 97.43%.

Figure 9: Continual learning scenarios detailed in Table 1 represented in a tree graph, as in [55].

For clarity we describe ABatchE for one linear layer, *i.e.* we describe the application of ABatchE to

$$f(\mathbf{x}, W) = \texttt{softmax}\left(W^\top \mathbf{x}\right) \qquad (9)$$

for input data $\mathbf{x} \in \mathbb{R}^m$ and weights $W \in \mathbb{R}^{m \times n}$. In BatchE [51], $W$ is trained on the first task then frozen. For task $i$ BatchE learns "fast weights" $r_i \in \mathbb{R}^m$, $s_i \in \mathbb{R}^n$ and outputs are computed via

$$f(\mathbf{x}, W) = \texttt{softmax}\left(\left(W \odot r_i s_i^\top\right)^\top \mathbf{x}\right). \qquad (10)$$

Wen *et al.* [51] further demonstrate that Equation 10 can be vectorized as

$$f(\mathbf{x}, W) = \texttt{softmax}\left(\left(W^\top \left(\mathbf{x} \odot r_i\right)\right) \odot s_i\right) \qquad (11)$$

or, for a batch of data $X \in \mathbb{R}^{b \times m}$,

$$f(X, W) = \texttt{softmax}\left(\left(\left(X \odot R_i^b\right) W\right) \odot S_i^b\right). \qquad (12)$$

In Equation 12, $R_i^b \in \mathbb{R}^{b \times m}$ is a matrix where each of the $b$ rows is $r_i$ (likewise $S_i^b \in \mathbb{R}^{b \times n}$ is a matrix where each of the $b$ rows is $s_i$).

As in Section 3.3 we now consider the case where data $X \in \mathbb{R}^{b \times m}$ comes from task $j$ but this information is not known to the model. For ABatchE we repeat the data $k$ times, where $k$ is the number of tasks learned so far, and use different "fast weights" for each repetiton. Specifically, we consider repeated data $\tilde{X} \in \mathbb{R}^{bk \times m}$ and augmented matricies $\tilde{R} \in \mathbb{R}^{bk \times m}$ and $\tilde{S} \in \mathbb{R}^{bk \times n}$ given by

$$\tilde{X} = \begin{bmatrix} X \\ X \\ \vdots \\ X \end{bmatrix}, \quad \tilde{R} = \begin{bmatrix} R_1^b \\ R_2^b \\ \vdots \\ R_k^b \end{bmatrix}, \quad \tilde{S} = \begin{bmatrix} S_1^b \\ S_2^b \\ \vdots \\ S_k^b \end{bmatrix}. \qquad (13)$$

Outputs are then computed as

$$f(X, W) = \texttt{softmax}\left(\left(\left(\tilde{X} \odot \tilde{R}\right) W\right) \odot \tilde{S}\right) \qquad (14)$$

Figure 10: Testing ABatchE on PermutedMNIST with LeNet 300-100 **(left)** and FC 1024-1024 **(right)** with output size 100.

where the $b$ rows $(bi, ..., bi + b - 1)$ of the output correspond exactly to Equation 12. The task may then be inferred by choosing the $i$ for which the rows $(bi, ..., b(i + 1) - 1)$ minimize the objective $\mathcal{H}$. If $f(X, W)_i$ denotes row $i$ of $f(X, W)$ then for objective $\mathcal{H}$ the inferred task for ABatchE is

$$\arg \min_i \sum_{\omega=0}^{b-1} \mathcal{H}\left(f(X, W)_{bi+\omega}\right). \tag{15}$$

To extend ABatchE to deep neural networks the matricies $\tilde{R}$ and $\tilde{S}$ are constructed for each layer.

One advantage of ABatchE over SupSup is that no backwards pass is required. However, ABatchE uses a very large batch size for large $k$, and the forward pass therefore requires more compute and memory. Another disadvantage of ABatchE is that the performance of ABatchE is limited by the performance of BatchE. In Section 4.2 we demonstrate that SupSup outperforms BatchE when BatchE is given task identity information.

Since the objective for ABatchE need not be differentiable we also experiment with an alternative metric of confidence $\mathcal{M}(\mathbf{p}) = -\max_i \mathbf{p}_i$. We showcase results for ABatchE on PermutedMNIST in Figure 10 for various values of $b$. The entropy objective $\mathcal{H}$ performs better than $\mathcal{M}$, and forgetting is only mitigated when using 16 images ($b = 16$). With 250 tasks, $b = 16$ corresponds to a batch size of 4000.

## D   Extended Training Details

### D.1   **SplitCIFAR-100** (GG)

As in [51] we train each model for 250 epochs per task. We use standard hyperparameters—the Adam optimizer [22] with a batch size of 128 and learning rate 0.001 (no warmup, cosine decay [28]). For SupSup we follow [39] and use non-affine normalization so there are no learned parameters. We do have to store the running mean and variance for each task, which we include in the parameter count. We found it better to use a higher learning rate (0.1) when training BatchE (Rand $W$), and the standard BatchE number is taken from [51].

### D.2   **SplitImageNet** (GG)

We use the Upper Bound and BatchE number from [51]. For SupSup we train for 100 epochs with a batch size of 256 using the Adam optimizer [22] with learning rate 0.001 (5 epochs warmup, cosine decay [28]). For SupSup we follow [39] and use non-affine normalization so there are no learned parameters. We do have to store the running mean and variance for each task, which we include in the parameter count.

### D.3   GNu **Experiments**

We clarify some experimental details for GNu experiments & baselines. For the BatchE [51] baseline we find it best to use kaiming normal initialization with a learning rate of 0.01 (0.0001 for the first task when the weights are trained). As we are considering hundreds of tasks, instead of training

Figure 11: **(left)** Interpolating between the binary and one-shot algorithm with $\gamma$. **(right)** Transfer enables faster learning on SplitCIFAR.

separate heads per tasks when training BatchE we also apply the rank one pertubation to the final layer. PSP [4] provides MNISTPerm results so we use the same hyperparameters as in their code. We compare with rotational superposition, the best performing model from PSP.

### D.4 Speed of the Masked Forward Pass

We now provide justification for the calculation mentioned in Section 4.1—when implemented properly the masking operation should require $\sim 1\%$ of the total time for a forward pass (for a ResNet-50 on a NVIDIA GTX 1080 Ti GPU). It is reasonable to assume that selecting indices is roughly as quick as memory access. A NVIDIA GTX 1080 Ti has a memory bandwidth of 480 GB/s. A ResNet-50 has around $2.5 \cdot 10^7$ 4-byte (32-bit) parameters—roughly 0.1 GB. Therefore, indexing over a ResNet-50 requires at most $0.1\,\text{GB}/\left(480\,\text{GB/s}\right) \approx 0.21$ ms. For comparison, the average forward pass of a ResNet-50 for a $3 \times 224 \times 224$ image on the same GPU is about 25 ms.

Note that NVIDIA hardware specifications generally assume best-case performance with sequential page reads. However, even if real-world memory bandwidth speeds are 60-70% slower than advertised, the fraction of masking time would remain in the $\leq 3\%$ range.

### D.5 Additional Transfer Experiment

For our transfer experiments, we initialize the score matrix (see Appendix E) for task $i$ with the running mean of the supermasks for tasks 0 through $i-1$. The scores for task 0 are initialized as in [39]. We further normalize by the Kaiming fan-in constant from [17], so that the norm of our supermask matrix is reasonable. If we do not perform this normalization, accuracy degrades significantly. All other training hyperparameters are the same as in Section D.1.

In Figure 11, we demonstrate that Transfer enables faster learning for SplitCIFAR. In this experiment, we train task 0 for the full 250 epochs and all subsequent tasks for either 50 epochs (with transfer) or 100 epochs (without transfer). We see that adding transfer yields an improvement even while using about half the number of training iterations overall.

## E  Supermask Training with Edge-Popup

For completeness we briefly recap the Edge-Popup algorithm for training supermasks as introduced by [39]. Consider a linear layer with inputs $\mathbf{x} \in \mathbb{R}^m$ and outputs $\mathbf{y} = (W \odot M)^\top \mathbf{x}$ where $W \in \mathbb{R}^{m \times n}$ are the fixed weights and $M \in \{0, 1\}^{m \times n}$ is the supermask. The Edge-Popup algorithm learns a score matrix $S \in \mathbb{R}_+^{m \times n}$ and computes the mask via $M = h(S)$. The function $h$ sets the top $k\%$ of entries in $S$ to 1 and the remaining to 0. Edge-Popup updates $S$ via the straight through estimator—$h$ is considered to be the identity on the backwards pass.

## F  Comparing Binary and One-Shot

In Figure 11 **(left)** we interpolate between the **Binary** and **One-Shot** algorithms. We replace line 6 of Algorithm 2, $g_i \leq$ **median**$(g)$, with $g_i \leq$ **top-**$\gamma$%**-element**$(g)$. Then when $\gamma = 1/2$ we recover the binary algorithm (as **median**$(g) =$ **top-**$50$%**-element**$(g)$) and when $\gamma = 1/k$ we recover the one-shot algorithm. A performance drop is observed from binary to one-shot for the difficult task of MNISTRotate—sequentially learning 36 rotations of MNIST (each rotation differing by 10 degrees).

## G  Tree Representation for the Continual Learning Scenarios

In Figure 9 the Continual Learning scenarios are represented as a tree. This resembles the formulation from [55] with some modifications, *i.e.* "Tasks share output head?" is replaced with "Tasks share labels" as it is possible to share the output head but not labels, *e.g.* SupSup in GNu.

## H  Corresponding Tables

In this section we provide tabular results for figures from Section 4.

Table 2: Accuracy on SplitCIFAR100 corresponding to Figure 2 **(right)**. SupSup with Transfer approaches the upper bound.

| Entry | Avg Acc@1 | Bytes |
|---|---|---|
| SupSup (GG) | $77.56 \pm 0.73$ | 408432 |
| SupSup (GG) | $83.62 \pm 0.74$ | 508432 |
| SupSup (GG) | $86.45 \pm 0.61$ | 695592 |
| SupSup (GG) | $88.09 \pm 0.64$ | 1035792 |
| SupSup (GG) | $89.06 \pm 0.75$ | 1630032 |
| SupSup (GG) | $89.57 \pm 0.64$ | 2487472 |
| SupSup (GG) Transfer | $79.53 \pm 1.31$ | 408432 |
| SupSup (GG) Transfer | $85.33 \pm 1.05$ | 508432 |
| SupSup (GG) Transfer | $88.52 \pm 0.85$ | 695592 |
| SupSup (GG) Transfer | $90.12 \pm 0.75$ | 1035792 |
| SupSup (GG) Transfer | $91.31 \pm 0.74$ | 1630032 |
| SupSup (GG) Transfer | $\textbf{91.66} \pm 0.74$ | 2487472 |
| BatchE (GG) | $79.75 \pm 1.00$ | 4640800 |
| BatchE (GG) - Rand $W$ | $74.96 \pm 0.68$ | 400240 |
| Separate Heads | $70.60 \pm 1.40$ | 4544560 |
| Separate Heads - Rand $W$ | $50.00 \pm 1.37$ | 184000 |
| Upper Bound | $91.62 \pm 0.89$ | 89675200 |

Table 3: Accuracy on PermutedMNIST with LeNet 300-100 corresponding to Figure 3 **(left)**.

| Entry | 10 | 50 | 100 | 150 | 200 | 250 | Avg |
|---|---|---|---|---|---|---|---|
| SupSup (GNu $\mathcal{H}$) | 93.65 | **93.68** | **93.68** | **93.66** | 93.64 | 93.62 | **93.66** |
| SupSup (GNu $\mathcal{G}$) | 93.69 | 93.67 | 93.67 | **93.66** | **93.65** | **93.63** | **93.66** |
| PSP (GG) | **94.80** | 83.58 | 64.62 | 51.18 | 42.69 | 36.74 | 62.27 |
| BatchE (GG) | 88.85 | 88.33 | 88.23 | 88.23 | 88.22 | 88.21 | 88.34 |
| Upper Bound | 94.94 | 95.01 | 94.99 | 94.95 | 94.91 | 94.86 | 94.94 |

Table 4: Accuracy on PermutedMNIST with FC 1024-1024 corresponding to Figure 3 **(right)**.

| Entry | 10 | 50 | 100 | 150 | 200 | 250 | Avg |
|---|---|---|---|---|---|---|---|
| SupSup (GNu $\mathcal{H}$) | 96.28 | 96.14 | 96.04 | 95.91 | 95.86 | 95.66 | 95.98 |
| SupSup (GNu $\mathcal{G}$) | 96.28 | **96.19** | **96.05** | **96.00** | **95.99** | **95.92** | **96.07** |
| PSP (GG) | **97.16** | 94.74 | 87.77 | 78.35 | 69.14 | 61.11 | 81.38 |
| BatchE (GG) | 92.84 | 92.40 | 92.36 | 92.34 | 92.33 | 92.32 | 92.43 |
| Upper Bound | 96.76 | 96.70 | 96.68 | 96.66 | 96.63 | 96.61 | 96.67 |

Table 5: Accuracy on PermutedMNIST with LeNet 300-100 corresponding to Figure 4.

| Entry | 500 | 1000 | 1500 | 2000 | 2500 | Avg |
|---|---|---|---|---|---|---|
| SupSup (GNu $\mathcal{H}$) | 93.49 | 93.47 | 93.46 | 93.45 | 93.45 | 93.46 |
| SupSup (GNu $\mathcal{G}$) | 93.49 | 93.48 | 93.46 | 93.45 | 93.45 | 93.47 |
| SupSup (NNs $\mathcal{H}$) | 93.49 | 93.46 | 93.46 | 93.45 | 92.54 | 93.28 |
| Upper Bound | 94.71 | 94.71 | 94.71 | 94.71 | 94.71 | 94.71 |

Table 6: Accuracy with FC 1024-1024 on RotatedMNIST corresponding to Figure 5 **(left)**.

| Entry | Avg |
|---|---|
| SupSup (GNu full batch $\mathcal{H}$) | **96.13** |
| BatchE (GG) | 92.40 |
| PSP (GG) | 95.87 |
| Lower Bound | 48.71 |
| Upper Bound | 98.01 |

Table 7: Accuracy with FC 1024-1024 on PermutedMNIST corresponding to Figure 5 **(right)**.

| Entry | 10 | 50 | 100 | 150 | 200 | 250 | Avg |
|---|---|---|---|---|---|---|---|
| SupSup (GNu $\mathcal{H}$) | **96.29** | **95.94** | **95.59** | **95.40** | **95.00** | **94.91** | **95.52** |
| BatchE (GNu full batch $\mathcal{H}$) | 91.94 | 91.90 | 92.04 | 92.04 | 92.04 | 92.04 | 92.00 |
| BatchE (GNu $\mathcal{H}$) | 66.08 | 61.89 | 60.93 | 59.33 | 57.37 | 55.74 | 60.22 |
| Upper Bound | 96.76 | 96.70 | 96.68 | 96.66 | 96.63 | 96.61 | 96.67 |

Table 8: Accuracy on PermutedMNIST with LeNet 300-100 corresponding to Figure 6 **(left)**.

| Entry | 10 | 50 | 100 | 150 | 200 | 250 | Avg |
|---|---|---|---|---|---|---|---|
| SupSup (GNu $s = 200$ $\mathcal{H}$) | 93.46 | 93.49 | 93.48 | 93.47 | 93.47 | 93.46 | 93.47 |
| SupSup (GNu $s = 200$ $\mathcal{G}$) | 93.46 | 93.48 | 93.47 | 93.47 | 93.47 | 93.46 | 93.47 |
| SupSup (GNu $s = 100$ $\mathcal{H}$) | 93.65 | **93.68** | **93.68** | **93.66** | 93.64 | 93.62 | **93.66** |
| SupSup (GNu $s = 100$ $\mathcal{G}$) | **93.69** | 93.67 | 93.67 | **93.66** | 93.65 | 93.63 | **93.66** |
| SupSup (GNu $s = 25$ $\mathcal{H}$) | 93.71 | 93.51 | 93.28 | 93.10 | 93.06 | 92.94 | 93.27 |
| SupSup (GNu $s = 25$ $\mathcal{G}$) | 93.83 | 93.66 | 93.60 | 93.48 | 93.43 | 93.36 | 93.56 |
| Lower Bound | 71.67 | 41.82 | 30.52 | 26.40 | 23.31 | 20.88 | 35.77 |
| Upper Bound | 94.94 | 95.01 | 94.99 | 94.95 | 94.91 | 94.86 | 94.94 |

Table 9: Accuracy on PermutedMNIST with FC 1024-1024 corresponding to Figure 6 (**right**).

| Entry | 10 | 50 | 100 | 150 | 200 | 250 | Avg |
|---|---|---|---|---|---|---|---|
| SupSup (GNu $s = 200$ $\mathcal{H}$) | 96.28 | 96.14 | 96.04 | 95.91 | 95.86 | 95.66 | 95.98 |
| SupSup (GNu $s = 200$ $\mathcal{G}$) | 96.28 | **96.19** | **96.05** | **96.00** | **95.99** | **95.92** | **96.07** |
| SupSup (GNu $s = 100$ $\mathcal{H}$) | 95.90 | 94.77 | 94.02 | 93.71 | 93.00 | 92.84 | 94.04 |
| SupSup (GNu $s = 100$ $\mathcal{G}$) | **96.31** | 95.83 | 95.60 | 95.32 | 95.05 | 94.88 | 95.50 |
| SupSup (GNu $s = 25$ $\mathcal{H}$) | 82.28 | 69.06 | 64.51 | 60.99 | 58.15 | 57.03 | 65.34 |
| SupSup (GNu $s = 25$ $\mathcal{G}$) | **96.31** | 93.17 | 91.20 | 90.26 | 89.04 | 88.19 | 91.36 |
| Lower Bound | 76.89 | 49.40 | 38.93 | 34.53 | 31.30 | 29.36 | 43.40 |
| Upper Bound | 96.76 | 96.70 | 96.68 | 96.66 | 96.63 | 96.61 | 96.67 |

# I  Analysis

In this section we assume a slightly more technical perspective. The aim is not to formally prove properties of the algorithm. Rather, we hope that a more mathematical language may prove useful in extending intuition. Just as the empirical work of [8, 57, 39] was given a formal treatment in [29], we hope for more theoretical work to follow.

Our grounding intuition remains from Section 3.3—the correct mask will produce the lowest entropy output. Moreover, since entropy is differentiable, gradient based optimization can be used to recover the correct mask. However, many questions remain: Why do superfluous neurons (Section 3.6) help? In the case of MNISTPermuation, why is a single gradient sufficient? Although it is a simple case, steps forward can be made by analyzing the training of a linear head on fixed features. With *random* features, training a linear head on fixed features is considered in the literature of reservoir computing [43], and more [1].

Consider $k$ different classification problems with fixed features $\phi(\mathbf{x}) \in \mathbb{R}^m$. Traditionally, one would use learned weights $W \in \mathbb{R}^{m \times n}$ to compute *logits*

$$\mathbf{y} = W^\top \phi(\mathbf{x}) \tag{16}$$

and output classification probabilities $\mathbf{p} = \texttt{softmax}(\mathbf{y})$ where

$$\mathbf{p}_v = \frac{\exp(\mathbf{y}_v)}{\sum_{v'=1}^n \exp(\mathbf{y}_{v'})}. \tag{17}$$

Recall that with SupSup we compute the *logits* for task $i$ using fixed random weights $W$ and a learned binary mask $M^i \in \{0, 1\}^{m \times n}$ as

$$\mathbf{y} = \left(W \odot M^i\right)^\top \phi(\mathbf{x}) \tag{18}$$

where $\odot$ denotes an element-wise product and no bias term is allowed. Moreover, $W_{uv} = \xi_{uv}\sqrt{2/m}$ where $\xi_{uv}$ is chosen independently to be either $-1$ or $1$ with equal probability and the constant $\sqrt{2/m}$ follows Kaiming initialization [17].

Say we are given data $\mathbf{x}$ from task $j$. From now on we will refer to task $j$ as the *correct* task. Recall from Section 3.3 that SupSup attempts to infer the *correct* task by using a weighted mixture of masks

$$\mathbf{y} = \left(W \odot \sum_i \alpha_i M^i\right)^\top \phi(\mathbf{x}) \tag{19}$$

where the coefficients $\alpha_i$ sum to one, and are initially set to $1/k$.

To infer the correct task we attempt to construct a function $\mathcal{G}(\mathbf{y}; \alpha)$ with the following property: For fixed data, $\mathcal{G}$ is minimized when $\alpha = \mathbf{e}_j$ ($\mathbf{e}_j$ denotes a $k$-length vector that is 1 in index $j$ and 0 otherwise). We can then infer the correct task by solving a minimization problem.

As in **One-Shot**, we use a single gradient computation to infer the task via

$$\arg\max_i \left(-\frac{\partial \mathcal{G}}{\partial \alpha_i}\right). \tag{20}$$

A series of Lemmas will reveal how a single gradient step may be sufficient when tasks are unrelated (*e.g.* as in PermutedMNIST). We begin with the construction of a useful function $\mathcal{G}$, which will correspond exactly to $\mathcal{G}$ in Section 3.6. As in Section 3.6, this construction is made possible through superfluous neurons (s-neurons): The true labels are in $\{1, ..., \ell\}$, and a typical output is therefore length $\ell$. However, we add $n - \ell$ s-neurons resulting in a vector $\mathbf{y}$ of length $n$.

Let $\mathbf{S}$ denote the set of s-neurons and $\mathbf{R}$ denote the set of *real* neurons where $|\mathbf{S}| = n - \ell$ and $|\mathbf{R}| = \ell$. Moreover, assume that a standard cross-entropy loss is used during training, which will encourage s-neurons to have small values.

**Lemma I.1.** *It is possible to construct a function $\mathcal{G}$ such that the gradient matches the gradient from the supervised training loss $\mathcal{L}$ for all s-neurons. Specifically, $\frac{\partial \mathcal{G}}{\partial y_v} = \frac{\partial \mathcal{L}}{\partial y_v}$ for all $v \in \mathbf{S}$ and 0 otherwise.*

*Proof.* Let $g_v = \frac{\partial \mathcal{G}}{\partial y_v}$. It is easy to ensure that $g_v = 0$ for all $v \notin \mathbf{S}$ with a modern neural network library like PyTorch [37] as *detaching*[4] the outputs from the neurons $v \notin \mathbf{S}$ prevents gradient signal from reaching them. In code, let y be the outputs and m be a binary vector with $\mathrm{m}_v = 1$ if $v \in \mathbf{S}$ and 0 otherwise, then

$$\mathtt{y\ =\ (1\ -\ m)\ *\ y.detach()\ +\ m\ *\ y} \tag{21}$$

will prevent gradient signal from reaching $\mathbf{y}_v$ for $v \notin \mathbf{S}$.

Recall that the standard cross-entropy loss is

$$\mathcal{L}(\mathbf{y}) = -\log\left(\frac{\exp(\mathbf{y}_c)}{\sum_{v'=1}^{n}\exp(\mathbf{y}_{v'})}\right) = -\mathbf{y}_c + \log\left(\sum_{v'=1}^{n}\exp(\mathbf{y}_{v'})\right) \tag{22}$$

where $c \in \{1, ..., \ell\}$ is the correct label. The gradient of $\mathcal{L}$ to any s-neuron $v$ is then

$$\frac{\partial \mathcal{L}}{\partial \mathbf{y}_v} = \frac{\exp(\mathbf{y}_v)}{\sum_{v'=1}^{n}\exp(\mathbf{y}_{v'})}. \tag{23}$$

If we define $\mathcal{G}$ as

$$\mathcal{G}(\mathbf{y}; \alpha) = \log\left(\sum_{v'=1}^{n}\exp(\mathbf{y}_{v'})\right) \tag{24}$$

then $g_v = \frac{\partial \mathcal{L}}{\partial y_v}$ as needed. Expressed in code

$$\mathtt{y\ =\ model(x);\ \ G\ =\ torch.logsumexp((1\ -\ m)\ *\ y.detach()\ +\ m\ *\ y,\ dim=1)} \tag{25}$$

where `model(...)` computes Equation 19. $\qquad\square$

In the next two Lemmas we aim to show that, in expectation, $-\frac{\partial \mathcal{G}}{\partial \alpha_i} \leq 0$ for $i \neq j$ while $-\frac{\partial \mathcal{G}}{\partial \alpha_j} > 0$. Recall that $j$ is the *correct* task—the task from which the data is drawn—and we will use $i$ to refer to a different task.

When we take expectation, it is with respect to the random variables $\xi, \{M^\omega\}_{\omega \in \{1,...,k\}}$, and $\mathbf{x}$. Before we proceed further a few assumptions are formalized, *e.g.* what it means for tasks to be unrelated.

**Assumption 1:** We assume that the mask learned on task $i$ will be independent from the data from task $j$: If the data is from task $j$ then $\phi(\mathbf{x})$ and $M^i$ and independent random variables.

**Assumption 2:** We assume that a negative weight and positive weight are equally likely to be masked out. As a result, $\mathbb{E}\left[\xi_{uv}M^i_{uv}\right] = 0$. Note that when $\mathbb{E}\left[\phi(\mathbf{x})\right] = 0$, which will be the case for zero mean random features, there should be little doubt that this assumption should hold.

**Lemma I.2.** *If data $\mathbf{x}$ comes from task $j$ and $i \neq j$ then*

$$\mathbb{E}\left[-\frac{\partial \mathcal{G}}{\partial \alpha_i}\right] \leq 0 \tag{26}$$

*Proof.* We may write the gradient as

$$\frac{\partial \mathcal{G}}{\partial \alpha_i} = \sum_{v=1}^{n} \frac{\partial \mathcal{G}}{\partial \mathbf{y}_v} \frac{\partial \mathbf{y}_v}{\partial \alpha_i} \tag{27}$$

and use that $\frac{\partial \mathcal{G}}{\partial \mathbf{y}_v} = 0$ for $v \notin \mathbf{S}$. Moreover, $\mathbf{y}_v$ may be written as

$$\mathbf{y}_v = \sum_{u=1}^{n} \phi(\mathbf{x})_u W_{uv} \left( \sum_{i=1}^{k} \alpha_i M_{uv}^i \right) \tag{28}$$

with $W_{uv} = \xi_{uv} \sqrt{2/m}$ and so Equation 27 becomes

$$\frac{\partial \mathcal{G}}{\partial \alpha_i} = \frac{\sqrt{2}}{\sqrt{m}} \sum_{v \in \mathbf{S}} \sum_{u=1}^{n} \frac{\partial \mathcal{G}}{\partial \mathbf{y}_v} \phi(\mathbf{x})_u \xi_{uv} M_{uv}^i. \tag{29}$$

Taking the expectation (and using linearity) we obtain

$$\mathbb{E}\left[\frac{\partial \mathcal{G}}{\partial \alpha_i}\right] = \frac{\sqrt{2}}{\sqrt{m}} \sum_{v \in \mathbf{S}} \sum_{u=1}^{n} \mathbb{E}\left[\frac{\partial \mathcal{G}}{\partial \mathbf{y}_v} \phi(\mathbf{x})_u \xi_{uv} M_{uv}^i\right]. \tag{30}$$

In Lemma J.1 we formally show that each term in this sum is greater than or equal to 0, which completes this proof. However, we can see informally now why expectation should be close to 0 if we ignore the gradient term as

$$\mathbb{E}\left[\phi(\mathbf{x})_u \xi_{uv} M_{uv}^i\right] = \mathbb{E}\left[\phi(\mathbf{x})_u\right] \mathbb{E}\left[\xi_{uv} M_{uv}^i\right] = 0 \tag{31}$$

where the first equality follows from Assumption 1 and the latter follows from Assumption 2. $\quad\square$

We have now seen that in expectation $-\frac{\partial \mathcal{G}}{\partial \alpha_i} \leq 0$ for $i \neq j$. It remains to be shown that we should expect $-\frac{\partial \mathcal{G}}{\partial \alpha_j} > 0$.

**Lemma I.3.** *If data* $\mathbf{x}$ *comes from the task* $j$ *then*

$$\mathbb{E}\left[-\frac{\partial \mathcal{G}}{\partial \alpha_j}\right] > 0. \tag{32}$$

*Proof.* Following Equation 30, it suffices to show that for $u \in \{1, ..., m\}$, $v \in \mathbf{S}$

$$\mathbb{E}\left[-\frac{\partial \mathcal{G}}{\partial \mathbf{y}_v} \phi(\mathbf{x})_u \xi_{uv} M_{uv}^j\right] > 0. \tag{33}$$

Since $v \in \mathbf{S}$ we may invoke Lemma I.1 to rewrite our objective as

$$\mathbb{E}\left[-\frac{\partial \mathcal{L}}{\partial \mathbf{y}_v} \phi(\mathbf{x})_u \xi_{uv} M_{uv}^j\right] > 0 \tag{34}$$

where $\mathcal{L}$ is the supervised loss used for training. Recall that in the mask training algorithm, real valued scores $S_{uv}^j$ are associated with $M_{uv}^j$ [39, 30]. The update rule for $S_{uv}^j$ on the backward pass is then

$$S_{uv}^j \leftarrow S_{uv}^j + \eta \left(-\frac{\partial \mathcal{L}}{\partial \mathbf{y}_v} \phi(\mathbf{x})_u \xi_{uv}\right) \tag{35}$$

for some learning rate $\eta$. Following Mallya *et al.* [30] (with threshold 0, as used in Section 4.2), we let $M_{uv}^j = 1$ if $S_{uv}^j > 0$ and otherwise assign $M_{uv}^j = 0$. As a result, we expect that $M_{uv}^j$ is 1 when $-\frac{\partial \mathcal{L}}{\partial \mathbf{y}_v} \phi(\mathbf{x})_u \xi_{uv}$ is more consistently positive than negative. In other words, the expected product of $M_{uv}^j$ and $-\frac{\partial \mathcal{L}}{\partial \mathbf{y}_v} \phi(\mathbf{x})_u \xi_{uv}$ is positive, satisfying Equation 34. $\quad\square$

Together, three Lemmas have demonstrated that in expectation $-\frac{\partial \mathcal{G}}{\partial \alpha_i} \leq 0$ for $i \neq j$ while $-\frac{\partial \mathcal{G}}{\partial \alpha_j} > 0$. Accordingly, we should expect that

$$\arg\max_i \left( -\frac{\partial \mathcal{G}}{\partial \alpha_i} \right). \tag{36}$$

returns the correct task $j$. While a full, formal treatment which includes the analysis of noise is beyond the scope of this work, we hope that this section has helped to further intuition. However, we are missing one final piece—what is the relation between $\mathcal{G}$ and $\mathcal{H}$?

It is not difficult to imagine that $\mathcal{H}$ should imitate the loss, which attempts to raise the score of one logit while bringing all others down. Analytically we find that $\mathcal{H}$ can be decomposed into two terms as follows

$$\mathcal{H}\left(\mathbf{p}\right) = -\sum_{v=1}^{n} \mathbf{p}_v \log \mathbf{p}_v \tag{37}$$

$$= -\sum_{v=1}^{n} \mathbf{p}_v \log \left( \frac{\exp\left(\mathbf{y}_v\right)}{\sum_{v'=1}^{n} \exp\left(\mathbf{y}_{v'}'\right)} \right) \tag{38}$$

$$= \left( -\sum_{v=1}^{n} \mathbf{p}_v \mathbf{y}_v \right) + \log \left( \sum_{v'=1}^{n} \exp\left(\mathbf{y}_{v'}'\right) \right) \tag{39}$$

where the latter term is $\mathcal{G}$. With more and more neurons in the output layer, $\mathbf{p}_v$ will become small moving $\mathcal{H}$ towards $\mathcal{G}$.

## J    Additional Technical Details

**Lemma J.1.** *If $j$ is the true task and $i \neq j$ then*

$$\mathbb{E}\left[ \frac{\partial \mathcal{G}}{\partial \mathbf{y}_v} \phi(\mathbf{x})_u \xi_{uv} M_{uv}^i \right] \geq 0 \tag{40}$$

*Proof.* Recall from Lemma I.1 that

$$\frac{\partial \mathcal{G}}{\partial \mathbf{y}_v} = \mathbf{p}_v = \frac{\exp(\mathbf{y}_v)}{\sum_{v'=1}^{n} \exp(\mathbf{y}_{v'})} \tag{41}$$

and so we rewrite equation 40 as

$$\mathbb{E}\left[ \mathbf{p}_v \phi(\mathbf{x})_u \xi_{uv} M_{uv}^i \right] \geq 0. \tag{42}$$

By the law of total expectation

$$\mathbb{E}\left[ \mathbf{p}_v \phi(\mathbf{x})_u \xi_{uv} M_{uv}^i \right] = \mathbb{E}\left[ \mathbb{E}\left[ \mathbf{p}_v \phi(\mathbf{x})_u \xi_{uv} M_{uv}^i \,\middle|\, \left| \phi(\mathbf{x})_u \xi_{uv} M_{uv}^i \right| \right] \right] \tag{43}$$

and so it suffices to show that

$$\mathbb{E}\left[ \mathbf{p}_v \phi(\mathbf{x})_u \xi_{uv} M_{uv}^i \,\middle|\, \left| \phi(\mathbf{x})_u \xi_{uv} M_{uv}^i \right| = \kappa \right] \geq 0 \tag{44}$$

for any $\kappa \geq 0$. In the case where where $\kappa = 0$ Equation 44 becomes

$$\mathbb{E}\left[ 0\mathbf{p}_v \,\middle|\, \left| \phi(\mathbf{x})_u \xi_{uv} M_{uv}^i \right| = 0 \right] = 0 \tag{45}$$

and so we are only left to consider $\kappa > 0$. Note that $\kappa > 0$ restricts $M_{uv}^i$ to be 1.

Again invoking the law of total expectation we rewrite Equation 45 as

$$\mathbb{E}\left[\mathbf{p}_v\phi(\mathbf{x})_u\xi_{uv}M_{uv}^i \middle| \left|\phi(\mathbf{x})_u\xi_{uv}M_{uv}^i\right|\right]$$

$$= \mathbb{E}\left[\mathbf{p}_v\phi(\mathbf{x})_u\xi_{uv}M_{uv}^i \middle| \phi(\mathbf{x})_u\xi_{uv}M_{uv}^i = \kappa\right]\mathbb{P}\left(\phi(\mathbf{x})_u\xi_{uv}M_{uv}^i = \kappa\right) \tag{46}$$

$$+ \mathbb{E}\left[\mathbf{p}_v\phi(\mathbf{x})_u\xi_{uv}M_{uv}^i \middle| \phi(\mathbf{x})_u\xi_{uv}M_{uv}^i = -\kappa\right]\mathbb{P}\left(\phi(\mathbf{x})_u\xi_{uv}M_{uv}^i = -\kappa\right).$$

Moreover, since the data is from task $j \neq i$, we can use Assumption 1 and 2 to show that each of the cases above is equally likely. Formally,

$$\mathbb{P}\left(\phi(\mathbf{x})_u\xi_{uv}M_{uv}^i = \kappa\right) \tag{47}$$

$$= \mathbb{P}\left(\left(\{\phi(\mathbf{x})_u = \kappa\} \cap \{\xi_{uv}M_{uv}^i = 1\}\right) \cup \left(\{\phi(\mathbf{x})_u = -\kappa\} \cap \{\xi_{uv}M_{uv}^i = -1\}\right)\right) \tag{48}$$

$$= \mathbb{P}\left(\phi(\mathbf{x})_u = \kappa\right)\mathbb{P}\left(\xi_{uv}M_{uv}^i = +1\right) + \mathbb{P}\left(\phi(\mathbf{x})_u = -\kappa\right)\mathbb{P}\left(\xi_{uv}M_{uv}^i = -1\right) \tag{49}$$

$$= \mathbb{P}\left(\phi(\mathbf{x})_u = \kappa\right)\mathbb{P}\left(\xi_{uv}M_{uv}^i = -1\right) + \mathbb{P}\left(\phi(\mathbf{x})_u = -\kappa\right)\mathbb{P}\left(\xi_{uv}M_{uv}^i = +1\right) \tag{50}$$

$$= \mathbb{P}\left(\left(\{\phi(\mathbf{x})_u = \kappa\} \cap \{\xi_{uv}M_{uv}^i = -1\}\right) \cup \left(\{\phi(\mathbf{x})_u = -\kappa\} \cap \{\xi_{uv}M_{uv}^i = +1\}\right)\right) \tag{51}$$

$$= \mathbb{P}\left(\phi(\mathbf{x})_u\xi_{uv}M_{uv}^i = -\kappa\right) \tag{52}$$

and so we may factor out the probability terms in Equation 46. Accordingly, it suffices to show that

$$\mathbb{E}\left[\mathbf{p}_v\phi(\mathbf{x})_u\xi_{uv}M_{uv}^i \middle| \phi(\mathbf{x})_u\xi_{uv}M_{uv}^i = \kappa\right] + \mathbb{E}\left[\mathbf{p}_v\phi(\mathbf{x})_u\xi_{uv}M_{uv}^i \middle| \phi(\mathbf{x})_u\xi_{uv}M_{uv}^i = -\kappa\right] \geq 0. \tag{53}$$

Before we proceed, we will introduce a function $h$ which we use to denote

$$h\left(\{\mathbf{y}_v\}, \kappa\right) = \kappa\frac{\exp(\mathbf{y}_v + \kappa)}{\exp(\mathbf{y}_v + \kappa) + \sum_{v' \neq v}\exp(\mathbf{y}_{v'})}. \tag{54}$$

for $\kappa > 0$. We will make use of two interesting properties of $h$.

We first note that $h\left(\{\mathbf{y}_v\}, \kappa\right) + h\left(\{\mathbf{y}_v\}, -\kappa\right) \geq 0$, which is formally shown in J.2.

Second, we note that

$$\mathbb{P}\left(\mathbf{p}_v\phi(\mathbf{x})_u\xi_{uv}M_{uv}^i = h\left(\{\mathbf{y}_v\}, \kappa\right) \middle| \phi(\mathbf{x})_u\xi_{uv}M_{uv}^i = \kappa\right)$$
$$= \mathbb{P}\left(\mathbf{p}_v\phi(\mathbf{x})_u\xi_{uv}M_{uv}^i = h\left(\{\mathbf{y}_v\}, -\kappa\right) \middle| \phi(\mathbf{x})_u\xi_{uv}M_{uv}^i = -\kappa\right) \tag{55}$$

which we dissect in Lemma J.3.

Utilizing these two properties of $h$ we may show that Equation 53 holds as

$$\mathbb{E}\left[\mathbf{p}_v\phi(\mathbf{x})_u\xi_{uv}M_{uv}^i \middle| \phi(\mathbf{x})_u\xi_{uv}M_{uv}^i = \kappa\right] + \mathbb{E}\left[\mathbf{p}_v\phi(\mathbf{x})_u\xi_{uv}M_{uv}^i \middle| \phi(\mathbf{x})_u\xi_{uv}M_{uv}^i = -\kappa\right] \tag{56}$$

$$= \int_{\mathbb{R}} h\left(\{\mathbf{y}_v\}, \kappa\right)\, d\mathbb{P}\left(\mathbf{p}_v\phi(\mathbf{x})_u\xi_{uv}M_{uv}^i = h\left(\{\mathbf{y}_v\}, \kappa\right) \middle| \phi(\mathbf{x})_u\xi_{uv}M_{uv}^i = \kappa\right)$$
$$+ \int_{\mathbb{R}} h\left(\{\mathbf{y}_v\}, -\kappa\right)\, d\mathbb{P}\left(\mathbf{p}_v\phi(\mathbf{x})_u\xi_{uv}M_{uv}^i = h\left(\{\mathbf{y}_v\}, -\kappa\right) \middle| \phi(\mathbf{x})_u\xi_{uv}M_{uv}^i = -\kappa\right) \tag{57}$$

$$= \int_{\mathbb{R}} \left(h\left(\{\mathbf{y}_v\}, \kappa\right) + h\left(\{\mathbf{y}_v\}, -\kappa\right)\right)\, d\mathbb{P}\left(\mathbf{p}_v\phi(\mathbf{x})_u\xi_{uv}M_{uv}^i = h\left(\{\mathbf{y}_v\}, \kappa\right) \middle| \phi(\mathbf{x})_u\xi_{uv}M_{uv}^i = \kappa\right) \tag{58}$$

$$\geq 0. \tag{59}$$

$\square$

**Lemma J.2.** $h\left(\{\mathbf{y}_v\}, \kappa\right) + h\left(\{\mathbf{y}_v\}, -\kappa\right) \geq 0.$

*Proof.* Recall that $\kappa \geq 0$. Moreover,

$$\exp(\mathbf{y}_v + \kappa) \sum_{v'} \exp(\mathbf{y}_{v'}) \geq \exp(\mathbf{y}_v - \kappa) \sum_{v'} \exp(\mathbf{y}_{v'}) \tag{60}$$

$$\Rightarrow \ \exp(\mathbf{y}_v + \kappa) \left( \exp(\mathbf{y}_v - \kappa) + \sum_{v'} \exp(\mathbf{y}_{v'}) \right)$$
$$\geq \exp(\mathbf{y}_v - \kappa) \left( \exp(\mathbf{y}_v + \kappa) + \sum_{v'} \exp(\mathbf{y}_{v'}) \right) \tag{61}$$

$$\Rightarrow \kappa \frac{\exp(\mathbf{y}_v + \kappa)}{\exp(\mathbf{y}_v + \kappa) + \sum_{v' \neq v} \exp(\mathbf{y}_{v'})} \geq \kappa \frac{\exp(\mathbf{y}_v - \kappa)}{\exp(\mathbf{y}_v - \kappa) + \sum_{v' \neq v} \exp(\mathbf{y}_{v'})} \tag{62}$$

and we may then subtract the term on the right from both sides. $\square$

**Lemma J.3.** *Consider take $i \neq j$ where $j$ is the correct task. Then*

$$\mathbb{P}\left( \mathbf{p}_v \phi(\mathbf{x})_u \xi_{uv} M_{uv}^i = h\left(\{\mathbf{y}_v\}, \kappa\right) | \phi(\mathbf{x})_u \xi_{uv} M_{uv}^i = \kappa \right)$$
$$= \mathbb{P}\left( \mathbf{p}_v \phi(\mathbf{x})_u \xi_{uv} M_{uv}^i = h\left(\{\mathbf{y}_v\}, -\kappa\right) | \phi(\mathbf{x})_u \xi_{uv} M_{uv}^i = -\kappa \right). \tag{63}$$

*Proof.* Note that this equation is satisfied when $\kappa = 0$ (since $-0 = 0$). For the remainder of this proof we will instead consider the case where $\kappa > 0$ (and so $M_{uv}^i = 1$).

If we define $\rho$ as $\rho = \left( \mathbb{P}\left( \phi(\mathbf{x})_u = \kappa \right) + \mathbb{P}\left( \phi(\mathbf{x})_u = -\kappa \right) \right)^{-1}$ then may decompose Equation 63 into four terms. Namely,

$$\mathbb{P}\left( \mathbf{p}_v \phi(\mathbf{x})_u \xi_{uv} M_{uv}^i = h\left(\{\mathbf{y}_v\}, \kappa\right) | \phi(\mathbf{x})_u = \kappa \right) \mathbb{P}\left( \phi(\mathbf{x})_u = \kappa \right) \rho$$
$$+ \mathbb{P}\left( \mathbf{p}_v \phi(\mathbf{x})_u \xi_{uv} M_{uv}^i = h\left(\{\mathbf{y}_v\}, \kappa\right) | \phi(\mathbf{x})_u = -\kappa \right) \mathbb{P}\left( \phi(\mathbf{x})_u = -\kappa \right) \rho$$
$$= \mathbb{P}\left( \mathbf{p}_v \phi(\mathbf{x})_u \xi_{uv} M_{uv}^i = h\left(\{\mathbf{y}_v\}, -\kappa\right) | \phi(\mathbf{x})_u = \kappa \right) \mathbb{P}\left( \phi(\mathbf{x})_u = \kappa \right) \rho$$
$$+ \mathbb{P}\left( \mathbf{p}_v \phi(\mathbf{x})_u \xi_{uv} M_{uv}^i = h\left(\{\mathbf{y}_v\}, -\kappa\right) | \phi(\mathbf{x})_u = -\kappa \right) \mathbb{P}\left( \phi(\mathbf{x})_u = -\kappa \right) \rho. \tag{64}$$

Equality follows from the fact that term 1 and 3 are equal, as are terms 2 and 4. We will consider terms 1 and 3, as the other case is nearly identical.

Let $H$ be the event where $\phi(\mathbf{x})_u = \kappa$, $M_{uv}^i = 1$ and all other random variables (except for $\xi_{uv}$) take values such that, if $\xi_{uv} = +1$ then $\mathbf{p}_v \phi(\mathbf{x})_u \xi_{uv} M_{uv}^i = h\left(\{\mathbf{y}_v\}, \kappa\right)$. On the other hand, if $\xi_{uv} = -1$ then $\mathbf{p}_v \phi(\mathbf{x})_u \xi_{uv} M_{uv}^i = h\left(\{\mathbf{y}_v\}, -\kappa\right)$. Then, subtracting term 3 from term 1 (and factoring out the shared term) we find

$$\mathbb{P}\left( \mathbf{p}_v \phi(\mathbf{x})_u \xi_{uv} M_{uv}^i = h\left(\{\mathbf{y}_v\}, \kappa\right) | \phi(\mathbf{x})_u = \kappa \right)$$
$$- \mathbb{P}\left( \mathbf{p}_v \phi(\mathbf{x})_u \xi_{uv} M_{uv}^i = h\left(\{\mathbf{y}_v\}, -\kappa\right) | \phi(\mathbf{x})_u = \kappa \right) \tag{65}$$
$$= \mathbb{P}\left( \xi_{uv} = +1 | H \right) - \mathbb{P}\left( \xi_{uv} = -1 | H \right) = 0 \tag{66}$$

since $\xi_{uv}$ is independent of $H$, and $\xi_{uv} = -1$ and $+1$ with equal probability. $\square$

## Footnotes

[4]`https://pytorch.org/docs/stable/autograd.html`