[Reviews · NeurIPS 2020]

Review 1

Summary and Contributions: This paper introduces a method, SupSup, for mitigating catastrophic forgetting that involves learning and applying task-specific binary weight masks to a fixed, randomly initialised network. In the scenario where task ids are available at both training and test time, SupSup is very similar to (and uses the same method to learn masks as) [1], but it uses a random network rather than a pretrained one. However, while some previous methods involving superposition of weight matrices (including [1]) require the task id to be provided at test time, SupSup can infer the task id using one of two efficient methods that work by identifying the mask that minimizes the output entropy of the network. The paper also introduces a method for inferring the introduction of a new task during training, to deal with scenarios where task ids are never provided. Further contributions include a method for efficiently storing activation masks in a Hopfield net, augmenting an existing method (BatchEnsemble) with one of their test-time task inference mechanisms, and showing that the use of “superfluous” output neurons can help task identification at test time. Empirically, SupSup is shown to outperform versions of two related baselines on long sequences of standard continual learning benchmarks in scenarios that vary according to whether the task ids are available at training or test time. When SupSup has to infer the task id at test time, it outperforms the baselines even when they are provided with the task id, in all but one experiments. [1] Mallya, Arun, Dillon Davis, and Svetlana Lazebnik. "Piggyback: Adapting a single network to multiple tasks by learning to mask weights." Proceedings of the European Conference on Computer Vision (ECCV). 2018.

Strengths: - The paper combines ideas from the study of subnetworks of random networks and from that of superposition of several models in one network to introduce a novel method for mitigating catastrophic forgetting. It also contributes a number of efficient techniques to allow the method to perform task inference - the one shot and binary methods, as well as the superfluous neurons. - The experiments demonstrate the effectiveness of the method when compared to two similar existing methods. The performance of SupSup over very long sequences of tasks is impressive (Figure 5). - The design of the model and the experiments emphasise the time and memory efficiency of the method, e.g.: - The time for task inference scales well with the number of tasks as it only requires one forward and backward pass through the weighted superposition of masks. - Figure 3 shows how SupSup can surpass the baselines in SplitCifar and SplitImagenet with much less memory storage.

Weaknesses: - The method for actually learning the supermasks is not explicitly described - instead the reader is referred to [1]. This should be explained in the appendix at the very least, even though it is stated that in theory that SupSup method does not focus on the specific way in which the supermasks are learned. - I could not find a direct empirical comparison of the Binary and OneShot methods for task inference. A clearer explanation is required for why one or the other is used for different experiments. - While interesting in its own right, the Hopfield net method seems out of place in the paper since (i) it involves masking the activations rather than the weights of the network, for efficiency reasons, but this makes it quite a different method to the original SupSup, and (ii) most of the explanation of how it works and all the experiments are in the appendix. - Number of baselines. While the method is compared in experiments to two related methods, it would have been useful to also include baselines from other classes of method that can theoretically operate in the same scenarios wrt to task id availability, such as those explicitly listed in Table 1, and particularly those that can infer task id at test time.

Correctness: Yes, as far as I am aware.

Clarity: Overall, the paper is well written and the motivations and claims are clearly outlined in the introduction. The paper introduces many ideas, including several variations of the SupSup model, all of which are valid contributions, but it results in a dense paper, when perhaps some extra features (e.g. HopSupSup) could have been skipped in favour of a clearer explanation of the background required to understand the model, e.g. how the supermasks are learnt and to include Figure 10 (currently in appendix) in the main paper, which clearly depicts the taxonomy of continual learning scenarios used in the paper.

Relation to Prior Work: Largely yes, related methods are described and SupSup is empirically compared to recent approaches in the same class of methods.

Reproducibility: Yes

Additional Feedback: *** POST-REBUTTAL COMMENTS *** Thank you to the authors for their response. I am glad to see an explicit comparison of the binary and one-shot methods, as well as more comparisons in the GNu setting, and I am looking forward to seeing a clear explanation of the supermask training algorithm in the final version. As a final comment, I am in agreement with reviewer 3 that the experiments section would be much stronger if you could demonstrate the performance of SupSup on more complex tasks in the GNu scenario, such as splitcifar and splitimagenet. *** - Line 123. “paramaterized” -> “parameterized” - Line 136. “paramaters” -> “parameters” - Section 3.2. It’s a bit confusing to mix results in with the methods, especially when you have to scroll all the way down to see Figure 3. - Line 163. “algorithms” -> “algorithm” - Line 198. “Hopfied” -> “Hopfield” - Section 3.4. In NN case, what if there is data in common between tasks, then gradient might be uniform and wrongly instantiate a new mask? Can you end up with many more supermasks than there are tasks for this reason? - Figure 3 table make clear that entries 2,3,4 refer to SupSup - bit confusing when you first see it. - Figure 5 is referenced before Figure 4. - Might be clearer to include Figure 10 in the main paper. - Would have been interesting to see empirical comparisons to [1] in GG setting, ie with a pretrained network. Even though it takes up more memory than SupSup, presumably this does not grow with the number of tasks? - Why are masks stored with 16 bit integers rather than single bits? - Why is PSP not a baseline for the splitcifar and splitimagenet GG experiments? - Have the authors thought about how it might be possible to extend this work to the continuous case? Ie no hard task boundaries?


Review 2

Summary and Contributions: In this paper, the authors propose a supermasks in superposition model for Continual Learning. If the task identity is given during tests, the correct subnetwork can be retrieved. If not provided, the proposed method infers the task using gradient-based optimization to find a linear superposition of the learned supermasks. The authors also give two extensions and perform experiments on PermutedMNIST, RotatedMNIST, and SplitMNIST, SplitCIFAR100, and SplitImageNet datasets.

Strengths: The authors design different scenarios for training and testing the proposed methods. The proposed method can tackle the situation where the task ID is not provided during inference. The proposed method can learn new tasks without forgetting previous tasks. The authors perform experiments on many continual learning datasets.

Weaknesses: This paper doesn’t cover previous continual learning approaches. It is hard to know how good the proposed method is without a sufficient comparison with previous methods. The novelty is limited. Learning supermasks for different tasks seems trivial to me. The design of different training and test scenarios, e.g. GNs and GNu, are not very common in previous continual learning literature. It's good to compare with previous methods under these scenarios. Some figures and tables are not very professional. The texts are either too big or too small.

Correctness: To my knowledge, the method is correct.

Clarity: The paper is well written.

Relation to Prior Work: I don't think this paper clearly discusses how it is different from previous contributions.

Reproducibility: Yes

Additional Feedback: After reading the feedback and the comments from other reviewers, I agree this paper has some merits, especially the ability of learning long sequences of tasks is important. But i still think the novelty of this paper is limited and the experiments are not comprehensive enough.


Review 3

Summary and Contributions: The paper extends work on the Lottery Ticket Hypothesis, which suggests that untrained networks masked appropriately can perform as well as their trained counterparts, to the continual learning scenario. By construction, the network cannot catastrophically forget, as the original weights are never modified. -> The authors propose a way to efficiently encode binary masks in order to reduce the memory burden of the approach. -> When task ID is unavailable at test time, the authors propose algorithms to infer the correct mask, based on the intuition that the correct mask wield yield low(er) entropy predictions -> Another criteria is proposed to infer the correct mask, by adding extra neurons which will be pushed down by the cross entropy objective -> The authors show strong results in the presented experiments -> The paper shows that keeping a running average of current masks can lead to good forward transfer

Strengths: This is an interesting paper, as it achieves strong performance (beating methods with task labels), while keeping the computational and memory cost low. The authors show that this method can indeed scale gracefully : they explore a very large number of tasks (for PermutedMNIST), compared to previous literature. Moreover, the method breaks away from the (n_task * n_targets_per_task) output size requirement in the GNu scenario, an important step to extending CL to very large sequences of tasks Applying supermasks to a continual learning scenario is definitely novel, and of interest to the community.

Weaknesses: My main criticism of the paper are the experiments done in the GNu scenario : while the results are impressive, they are limited to variants of MNIST. It's unclear if the proposed method can really work in more realistic / challenging scenarios (such as CIFAR or (mini)Imagenet). I would clarify this point in the introduction Moreover, a key property of CL methods is forward transfer : a good algorithm should enable faster learning of newer tasks. The paper shows some results for this (in "transfer"), however the improvement is somewhat minimal. Could the authors comment on this ? I'm interested to know if learning with transfer enables faster learning on new tasks (i.e. less epochs are required to reach X% of final accuracy)

Correctness: Claims, method, methodology is correct.

Clarity: Paper is well written, and in general easy to follow. There are some figures which are explained somewhat far from the figure location, which ideally should be fixed in a later version. Also, when referring to the appendix, please say explicitly the work appendix instead of just the section

Relation to Prior Work: The paper has a lot of prior work to cover (CL, supermasks, batch ensembles) and is done correctly. Prior work is concise and clear.

Reproducibility: Yes

Additional Feedback: In general I think this approach is an interesting take on the continual learning problem. The general approach and the methodology should be of interest to the CL community. While the results are interesting, I feel like the paper is somewhat disconnected from real applications (where you need to do task inference on "real" images, or that you want to reduce training costs by forward transfer). I'm ready to bump up my score if the authors : 1) can show convincingly that the transfer approach can have a meaningful impact on computational cost. An interesting experiment would be to compare the test accuracy after few (e.g. only one) epochs of training for transfer vs no transfer in figure 3 right, or 2) show that task inference can be done on more challenging datasets (anything other than mnist, really) *** post rebuttal edit *** Thank you for addressing the concerns I raised in your rebuttal. 1) The transfer experiment is a good start. If I may, the paper would make for an even stronger submission if you could add baselines like BatchE to that figure. 2) I'm somewhat underwhelmed with the permuted CIFAR-10 experiment. What I wanted to see was whether or not your method can scale to natural images, and permuting CIFAR pixels and training a dense network does not accomplish this. With that being said, I still think the proposed approach opens a new door of research for CL, so I'm updating my score to 7.

[Author Response · NeurIPS 2020]



Figure 1: **(left)** Interpolating between the binary and one-shot algorithm with $\gamma$. **(center)** Transfer enables faster learning on SplitCIFAR. **(right)** One-shot vs. binary for permutations of CIFAR. Figures viewed best with zoom.

• **R1**: Thank you for the great suggestions and thorough review, we look forward to fully incorporating your detailed
recommendations to improve the paper.

**Empirical comparison between binary and one-shot.** An empirical comparison between the binary and one-shot
algorithms is a fantastic addition: In Figure 1 **(left)** we directly interpolate between these two algorithms. We replace
line 6 of the binary algorithm, $g_i \leq \mathbf{median}(g)$, with $g_i \leq \mathbf{top}\text{-}\gamma\%\text{-}\mathbf{element}(g)$. Then when $\gamma = 1/2$ we recover the
binary algorithm (as $\mathbf{median}(g) = \mathbf{top}\text{-}50\%\text{-}\mathbf{element}(g)$) and when $\gamma = 1/k$ we recover the one-shot algorithm. A
performance drop is observed from binary to one-shot for the difficult task of MNISTRotate—sequentially learning 36
rotations of MNIST (each new rotation differing by only 10 degrees).

**Further comparison in** GNu**.** We believe the comparison of SupSup (in GNu) with recent methods (PSP [1], BatchE
[2]) in the GG scenario is fair since GG is strictly easier than GNu. However we agree that this is a weakness and will
update the paper to compare SupSup with methods *e.g.* from [3]. The initial reason for comparison of SupSup in GNu
with recent methods in the strictly easier GG scenario is because they were more competitive. For instance [3] considers
sequential learning problems with only 5-10 tasks. SupSup, after sequentially learning 250 permutations of MNIST,
outperforms all non-replay methods from [3] in the GNu scenario after they have learned only 10 permutations of
MNIST with a similar network: In GNu, Online EWC achieves 33.88% & SI achieves 29.31% on 10 permutations of
MNIST [3] while SupSup achieves 94.91% accuracy after 250 permutations (see Table 5 in [3] vs. Table 7 in our work).

**Additional comments.** We have updated the appendix to explicitly detail the supermask training algorithm, which
improves clarity. The method provided for NNs will not work with data that is common between tasks. We say that 16
bit integers are used instead of single bits because they store the index of the nonzero elements of the mask with the
CSC sparse matrix format. PSP on SplitCIFAR achieves worse performance than BatchE (GG) with similar bytes. We
will definitely think about extensions of SupSup to the continuous case, but do not currently have a solution.

• **R2**: We appreciate the suggestions, in particular to enhance the clarity of the figures which will improve the paper. For
further comparisons please see the **Further Comparison in** GNu section in **R1** above. In reference to the comments
concerning lack of novelty and lack of coverage of previous approaches we highlight **R3**'s comments: 1) *Applying*
*supermasks to a continual learning scenario is definitely novel, and of interest to the community* and 2) *The paper has a*
*lot of prior work to cover (CL, supermasks, batch ensembles) and is done correctly. Prior work is concise and clear.*
The most similar approach to SupSup is [4] and they are limited to scenario GG while requiring more storage.

• **R3**: We are grateful for a comprehensive and thoughtful review. We complete variants of the two very useful
experiments that you have suggested, and detail the results below.

**Forward transfer.** Thank you for highlighting the importance of transfer. We illustrate in Figure 1 **(center)** that our
method for transfer (initializing each new mask with a running mean of previous masks) does enable faster learning
(less epochs are required to reach a given accuracy). Training all tasks for 50 epochs with our transfer method provides
a significant accuracy boost over training individually on all tasks for 100 epochs for SplitCIFAR.

**Towards more complex tasks.** We illustrate in Figure 1 **(right)** that SupSup with the binary algorithm can sequentially
learn 50 permutations of CIFAR pixels with minimal forgetting. As in the paper we use the FC-1024-1024 network
and train on each task for 1000 iterations, and accordingly the upper bound accuracy is low. However, we are most
interested in the deviation from the upper bound. We will update the introduction to better reflect the scope of the paper.

[1] Brian Cheung *et al.*. Superposition of many models into one. NeurIPS 2019.

[2] Yeming Wen, Dustin Tran, and Jimmy Ba. Batchensemble: an alternative approach to efficient ensemble and lifelong learning.
ICLR 2020.

[3] Gido M van de Ven and Andreas S Tolias. Three scenarios for continual learning. arXiv, 2019.

[4] Arun Mallya, *et al.*: Adapting a single network to multiple tasks by learning to mask weights. ECCV, 2018.


[Meta-Review · NeurIPS 2020]

Reviewers agreed that this work makes an interesting and solid contribution, and of great interest to the community. While there was some debate on the benchmarks used, the reviewers concluded that the reported performance is strong; additional experiments could further demonstrate the effectiveness of the approach.